# Human formin FHOD3-mediated actin elongation is required for sarcomere integrity in cardiomyocytes

**Dylan A Valencia[1], Angela N Koeberlein[1], Haruko Nakano[2,3], Akos Rudas[4], Aanand A Patel[1†], Airi Harui[5], Cassandra Spencer[2], Atsushi Nakano[2,3,6,7]\*, Margot E Quinlan[1,6]\***

[1]Department of Chemistry and Biochemistry, University of California, Los Angeles, Los Angeles, United States; [2]Department of Molecular, Cell, and Developmental Biology, University of California, Los Angeles, Los Angeles, United States; [3]Eli and Edythe Broad Center of Regenerative Medicine and Stem Cell Research, University of California, Los Angeles, Los Angeles, United States; [4]Department of Computational Medicine, University of California, Los Angeles, Los Angeles, United States; [5]Department of Medicine, Divison of Pulmonary and Critical Care Medicine, Geffen School of Medicine, University of California, Los Angeles, Los Angeles, United States; [6]Molecular Biology Institute, University of California, Los Angeles, Los Angeles, United States; [7]Department of Molecular Physiology, The Jikei University School of Medicine, Tokyo, Japan

**\*For correspondence:**
anakano@ucla.edu (AN);
margot@chem.ucla.edu (MEQ)

**Present address:** †Department of Laboratory Medicine and Pathology, University of Washington, Seattle, United States

**Competing interest:** The authors declare that no competing interests exist.

## eLife Assessment

Valencia et al. combine elegant in vitro biochemical experiments with functional assays in cardiomyocytes to determine which properties of the FHOD3 formin are essential for sarcomere assembly. Using separation-of-function mutants, they show that FHOD3's elongation activity, rather than its nucleation, capping, or bundling activities, is key to its sarcomeric function. This is an **important** finding and the data presented in the manuscript are **convincing**; however, the presence of FHOD3 at filament barbed ends in the TIRF elongation assays should probably be verified directly in a future study.

**Abstract** Contractility and cell motility depend on accurately controlled assembly of the actin cytoskeleton. Formins are a large group of actin assembly proteins that nucleate and elongate new actin filaments. Some formins may cap filaments while others sever or bundle filaments. The formin homology domain-containing protein (FHOD) family of formins is critical to the formation of the fundamental contractile unit in muscle, the sarcomere. Specifically, mammalian FHOD3L plays an essential role in cardiomyocytes. Despite our knowledge of FHOD3L's importance in cardiomyocytes, its biochemical and cellular activities remain poorly understood. It was proposed that FHOD-family formins act by capping and bundling, as opposed to assembling new filaments. Here, we demonstrate that human FHOD3L nucleates actin and rapidly but briefly elongates filaments after temporarily pausing elongation. We designed function-separating mutants that enabled us to distinguish which biochemical roles are required in the cell. We found that FHOD3L's elongation activity, but not its nucleation, capping, or bundling activity, is necessary for proper sarcomere formation and contractile function in neonatal rat ventricular myocytes. The results of this work provide new insight into the mechanisms by which formins build specific structures and will contribute to knowledge regarding how cardiomyopathies arise from defects in sarcomere formation and maintenance.

## Introduction

Higher-order actin-based structures, such as the sarcomere, are the foundation for specific cellular functions that demand precise spatial and temporal coordination. Other examples of actin-based structures, including filopodia, stress fibers, and the cytokinetic furrow, are required for a wide range of processes, such as cell migration, adhesion, and division (*Blanchoin et al., 2014*; *Svitkina, 2018*). Hundreds of actin-binding proteins, and specific combinations thereof, coordinate to assemble these complex structures by accelerating actin assembly, cross-linking and bundling filaments, and stabilizing and/or disassembling actin filaments (*Dominguez and Holmes, 2011*; *Lappalainen et al., 2022*; *Pollard, 2016*). There are three known classes of actin nucleators that stimulate de novo filament assembly. Each class functions by a distinct mechanism, resulting in a unique starting point for the structure it initiates. One such class of actin nucleators, known as formins, mediates both nucleation and continued elongation of actin filaments. Formins are defined by their formin homology (FH) domains 1 and 2. The FH2 domain is highly conserved and forms a donut-shaped homodimer, which is sufficient to nucleate new filaments. After nucleation, the FH2 domain remains processively associated with the faster-growing barbed end of actin filaments, modifying the rate of elongation. The FH1 domain is less well conserved but contains proline-rich tracts that bind profilin-actin to facilitate elongation of actin filaments (*Courtemanche, 2018*; *Paul and Pollard, 2008*; *Paul and Pollard, 2009*). The regulatory domains for most formins are on either side of these FH domains, and an intramolecular interaction leads to autoinhibition (*Goode and Eck, 2007*; *Li and Higgs, 2005*; *Schönichen and Geyer, 2010*). The timing and strength of formin activity differ between formins, allowing for distinct roles within a given cell (*Homa et al., 2021*).

Multiple classes of formins have been linked to sarcomere structure, including disheveled associated activator of morphogenesis (Daam1, 2), diaphanous (Diaph1-3), formin homology domain-containing proteins (FHOD1, 3), and others (*Deng et al., 2021*; *Iskratsch et al., 2010*; *Li et al., 2011*; *Mi-Mi et al., 2012*; *Molnár et al., 2014*; *Rosado et al., 2014*; *Sanger et al., 2017*; *Shwartz et al., 2016*; *Sundaramurthy et al., 2020*; *Taniguchi et al., 2009*). There are two mammalian isoforms in the FHOD family of formins, FHOD1 and FHOD3 (*Bechtold et al., 2014*; *Kanaya et al., 2005*; *Schönichen and Geyer, 2010*). FHOD1 is widely expressed and assembles stress fibers that contribute to adhesion and motility of various cell types (*Gasteier et al., 2003*; *Iskratsch et al., 2013*; *Koka et al., 2003*; *Schulze et al., 2014*). Despite its localization to intercalated discs, structures important for rapid signal transmission and synchronized contractions, FHOD1 seems to be dispensable in cardiomyocytes (*Al Haj*

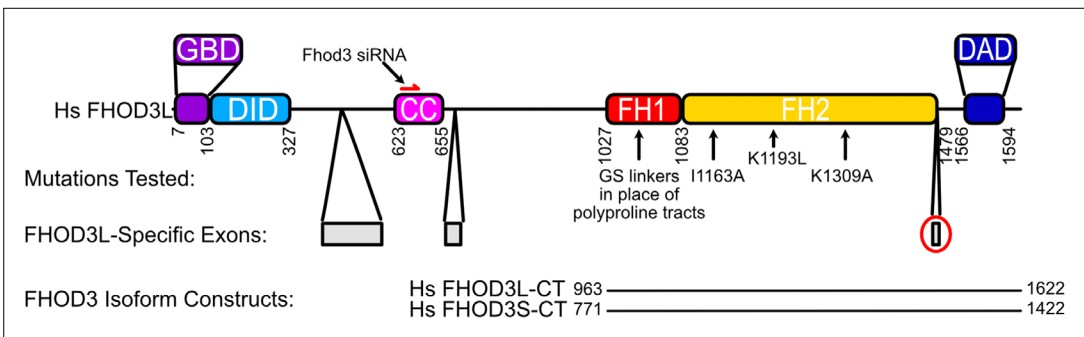

**Figure 1.** Domain structure of human FHOD3. GBD = GTPase binding domain, DID = diaphanous inhibitory domain, CC = coiled coil (putative), FH = formin homology, DAD = diaphanous autoregulatory domain. Mutations tested in this study are indicated with black arrows. Numbers correspond to the FHOD3L sequence (Uniprot isoform 4). The FHOD3L-CT construct spans residues 963–1622, including the FH1 domain, FH2 domain, and tail. The FHOD3S-CT construct spans residues 771–1422 (Uniprot isoform 1 numbering). FHOD3L-specific exons are in gray. The eight-residue exon (T(D/E)$_5$XE) that distinguishes FHOD3L-CT from FHOD3S-CT is circled in red.

The online version of this article includes the following source data and figure supplement(s) for figure 1:

**Figure supplement 1.** Coomassie-stained polyacrylamide gel of purified FHOD3 constructs.

**Figure supplement 1—source data 1.** PDF file containing original file for Coomasie-stained SDS-PAGE gel displayed in *Figure 1—figure supplement 1*, indicating the relevant bands and treatments.

**Figure supplement 1—source data 2.** Original file for Coomasie-stained SDS-PAGE gel displayed in *Figure 1—figure supplement 1*.

*et al., 2015*; *Dwyer et al., 2014*; *Sanematsu et al., 2019*). In contrast, FHOD3 is implicated in sarcomere assembly and maintenance (*Fujimoto et al., 2016*; *Iskratsch et al., 2010*; *Kan et al., 2012a*; *Taniguchi et al., 2009*; *Ushijima et al., 2018*).

There are four isoforms of FHOD3 reported in Uniprot. Major differences are due to alternative splicing in the N-terminal half. An eight-residue insert in the C-terminal half, an acidic sequence (T(D/E)$_5$XE), is present only in Uniprot isoform 4, which is commonly referred to as FHOD3L (*Iskratsch et al., 2010*). In this paper, we adopt the commonly accepted nomenclature in which FHOD3S (short) indicates Uniprot isoform 1, and FHOD3L (long) indicates Uniprot isoform 4 (*Figure 1*). FHOD3L is predominantly expressed in striated muscle, whereas FHOD3S is more widely expressed (*Iskratsch et al., 2010*; *Taniguchi et al., 2009*). In cardiomyocytes, FHOD3L localizes to sarcomeres in a striated pattern (*Iskratsch et al., 2010*; *Kan et al., 2012b*; *Taniguchi et al., 2009*). Interestingly, FHOD3L does not localize at the barbed end of actin filaments within sarcomeres. Instead, a direct interaction between FHOD3L and cardiac myosin-binding protein C (cMyBP-C) drives localization of FHOD3L to the so-called C zone of sarcomeres (*Matsuyama et al., 2018*). In the absence of cMyBP-C, FHOD3L is diffuse and cardiac function is compromised, suggesting that the balance between these two proteins is critical for proper heart function in mice (*Matsuyama et al., 2018*).

Mutations in the *FHOD3* gene are deemed causative in at least 1–2% of patients with hypertrophic cardiomyopathy (HCM), in addition to cases of left ventricular noncompaction, dilated cardiomyopathy, and progressive high-frequency hearing loss (*Arimura et al., 2013*; *Boussaty et al., 2023*; *Myasnikov et al., 2022*; *Ochoa, 2018*). Moreover, recent clinical studies show that HCM-linked FHOD3 mutations increase the risk of cardiovascular death and all-cause death, with the onset of the disease occurring as early as age 4 and as late as age 63 (*Vodnjov et al., 2023*; *Wu et al., 2021*).

Despite its known physiological significance, we lack a mechanistic understanding of FHOD3L's role in cardiac development and function. Early biochemical analysis suggested that FHOD-family formins were an atypical class of formins. Both purified mammalian isoforms (FHOD1 and FHOD3) were found to decelerate, rather than accelerate, actin assembly in vitro (*Schönichen et al., 2013*; *Taniguchi et al., 2009*). In addition, actin bundling was observed for FHOD1 (*Schönichen et al., 2013*). Consistently, FHOD1 mediates nuclear movement by bundling and anchoring actin filaments (*Antoku et al., 2023*). However, it fails to complete this function when a highly conserved isoleucine in the FH2 domain is mutated (hereafter, referred to as the IA mutation) (*Kutscheidt et al., 2014*). The IA mutation is known to disrupt both nucleation and elongation in most, if not all, formins, indicating that FHOD1 may enhance actin assembly in vivo (*Patel et al., 2018*; *Xu et al., 2004*). In addition, FHOD proteins are required to form new sarcomeres in rat cardiomyocytes, human induced cardiomyocytes, worms, and mice (*Iskratsch et al., 2010*; *Kan et al., 2012a*; *Mi-Mi et al., 2012*; *Taniguchi et al., 2009*). FHODs with the IA mutation do not support the formation of new sarcomeres, suggesting that actin assembly activity is required for this process, in conflict with the biochemical data (*Kan et al., 2012a*; *Shwartz et al., 2016*; *Taniguchi et al., 2009*; *Xu et al., 2004*). Most recently, in the worm, FHOD protein was shown to cooperate with profilin to build muscle, providing strong evidence that it functions by elongating actin filaments (*Kimmich et al., 2024*). Previously, we found that *Drosophila* Fhod is a potent actin nucleator that can accelerate actin elongation under certain conditions (*Bremer et al., 2024*; *Patel et al., 2018*). We also showed that human FHOD1 can nucleate actin, but its activity is sensitive to the actin isoform (*Patel et al., 2018*). We confirmed that both of these Fhodfamily formins lose activity when the IA mutation is introduced (*Patel et al., 2018*). Based on these biochemical results and the data regarding Fhod-family formins in muscle of multiple species, we asked whether human FHOD3L can also nucleate.

Here, we show that purified FHOD3L can accelerate actin assembly by both nucleation and elongation. We also confirm that FHOD3L potently caps filaments, in the absence of profilin, and bundles. Using function-separating mutations, we correlate FHOD3L's biochemistry to the integrity of the sarcomere within cardiomyocytes. We found that reduced nucleation, capping, and bundling by FHOD3L are well tolerated in neonatal rat ventricular myocytes (NRVMs), whereas elongation activity is necessary for proper sarcomere formation and function.

## Results

### Biochemical characterization of human FHOD3

We purified the C-terminal half of human FHOD3L (FHOD3L-CT), encompassing the FH1 domain, FH2 domain, and C-terminal tail, which is typically sufficient for actin assembly in vitro and in vivo (*Figure 1*, *Figure 1—figure supplement 1*; *Courtemanche, 2018*; *Patel et al., 2018*). In contrast to an earlier report, we found that FHOD3L-CT enhances rabbit skeletal muscle actin (RSA) assembly in bulk pyrene assays (*Figure 2A and B*, *Table 1*; *Taniguchi et al., 2009*). (*Table 1* summarizes all biochemical measurements.) Consistent with nucleation activity, when we visualized the products of similar reactions, we observed many more filaments in the presence of FHOD3L-CT compared to actin alone (*Figure 2—figure supplement 1A*). We also asked if the shorter, alternatively spliced FHOD3 isoform, FHOD3S-CT, nucleates actin. FHOD3S-CT, which lacks eight acidic residues at the end of its FH2 domain compared to FHOD3L-CT, accelerates actin assembly (~30%) more potently than FHOD3L-CT (*Figure 2B*, *Figure 2—figure supplement 2A*). We quantified the activity of each construct by plotting the slope of the actin assembly curve shortly after the reaction is initiated ($t_{1/8}$ = time until reaction has reached 1/8th completion) as a function of the FHOD3L-CT added. The slopes of these data points are then used to compare specific activity levels. In the cases of FHOD3L-CT vs FHOD3S-CT, the slopes are 0.32 ±0.04 vs 0.42 ± 0.04, respectively. Thus, we conclude that the C-terminal half of both FHOD3 isoforms (hereafter, FHOD3S/L-CT) is able to nucleate actin filaments.

To further compare FHOD3L-CT to previously characterized formins, we introduced mutations at conserved residues of the FH2 domain, I1163A and K1309A (*Xu et al., 2004*). Like other formins, the I1163A mutant lacked nucleation activity, while the K1309A mutant reduced nucleation by ~85% (*Figure 2B*). We observed a reduction in the plateau of pyrene traces with wild-type FHOD3S/L-CT and the mutants. The decrease was more apparent at higher concentrations of the nucleator and seemed to be dose-dependent (*Figure 2A*, *Figure 2—figure supplement 2A*). Control experiments demonstrated that the fluorescence change does not reflect quenching due to side binding or bundling (*Figure 2—figure supplement 1B*). We, therefore, measured barbed-end binding. We asked whether FHOD3L-CT inhibits filament elongation by performing bulk seeded elongation assays. Indeed, we found that FHOD3L-CT potently slows barbed-end elongation ($K_{app}$ = 0.023 ±0.005 nM; *Figure 2C and D*). (The shapes of these traces reflect the presence of activity in addition to capping [perhaps nucleation or bundling], especially at high concentrations, which could affect our estimate of the binding affinity.) Barbed-end binding by FHOD3S-CT is over 30 times weaker ($K_{app}$ = 0.750 ±0.090 nM) (*Figure 2D*). The IA mutation is generally thought to disrupt binding to both actin monomers and filament barbed ends. However, FHOD3L-CT I1163A has an affinity of 4.9 ±1.4 nM for barbed ends of actin filaments (*Figure 2D*, *Figure 2—figure supplement 2C*). While the affinity is >200-fold weaker than FHOD3L-CT, it still binds barbed ends tightly. Therefore, we interpret the plateau decrease observed in bulk assembly assays as evidence of barbed-end capping.

To directly observe the impact of FHOD3L-CT on growing actin filaments, we used total internal reflection fluorescence (TIRF) microscopy in the presence of profilin. Initially, we could not confirm that FHOD3L-CT was bound to these filaments. Most of the filaments elongated at rates indistinguishable from actin alone. We, therefore, repeated bulk seeded elongation assays in the presence of profilin. Under these conditions, we observed increased rates of actin assembly at low doses of FHOD3L-CT, confirming that the formin accelerates elongation (*Figure 2E*). In contrast, at higher formin concentrations, we found that elongation was actually slower than actin alone. We attribute the complicated shapes and dose response of these data to the contributions of elongation and capping. We also note that the concentrations required for FHOD3L-CT to elicit measurable changes in actin assembly indicate a marked apparent decrease (500- to 1000-fold) in affinity for the barbed end when profilin is present (*Figure 2E*).

Upon closer observation of individual filaments in the TIRF assay, we detected short dim stretches along the filaments. The dim regions reflected 'bursts' of fast elongation (*Figure 2F*). During the bursts, the filaments elongated ~3-fold faster (39 ± 13 subunits/s), consistent with formin-mediated elongation (*Figure 2F and H*, *Figure 2—figure supplement 1D*, *Figure 2—video 1*). The average length of filament produced during these bursts was 1.10 ±0.49 µm (*Figure 2—figure supplement 1E*), which is similar to the characteristic run length of *Drosophila* Fhod-A (~2 µm), but very different from other formins, which often range from 20 to 200 µm (*Bremer et al., 2024*; *Cao et al., 2018*; *Moseley and Goode, 2005*; *Patel et al., 2018*; *Vizcarra et al., 2014*). Interestingly, a pause often

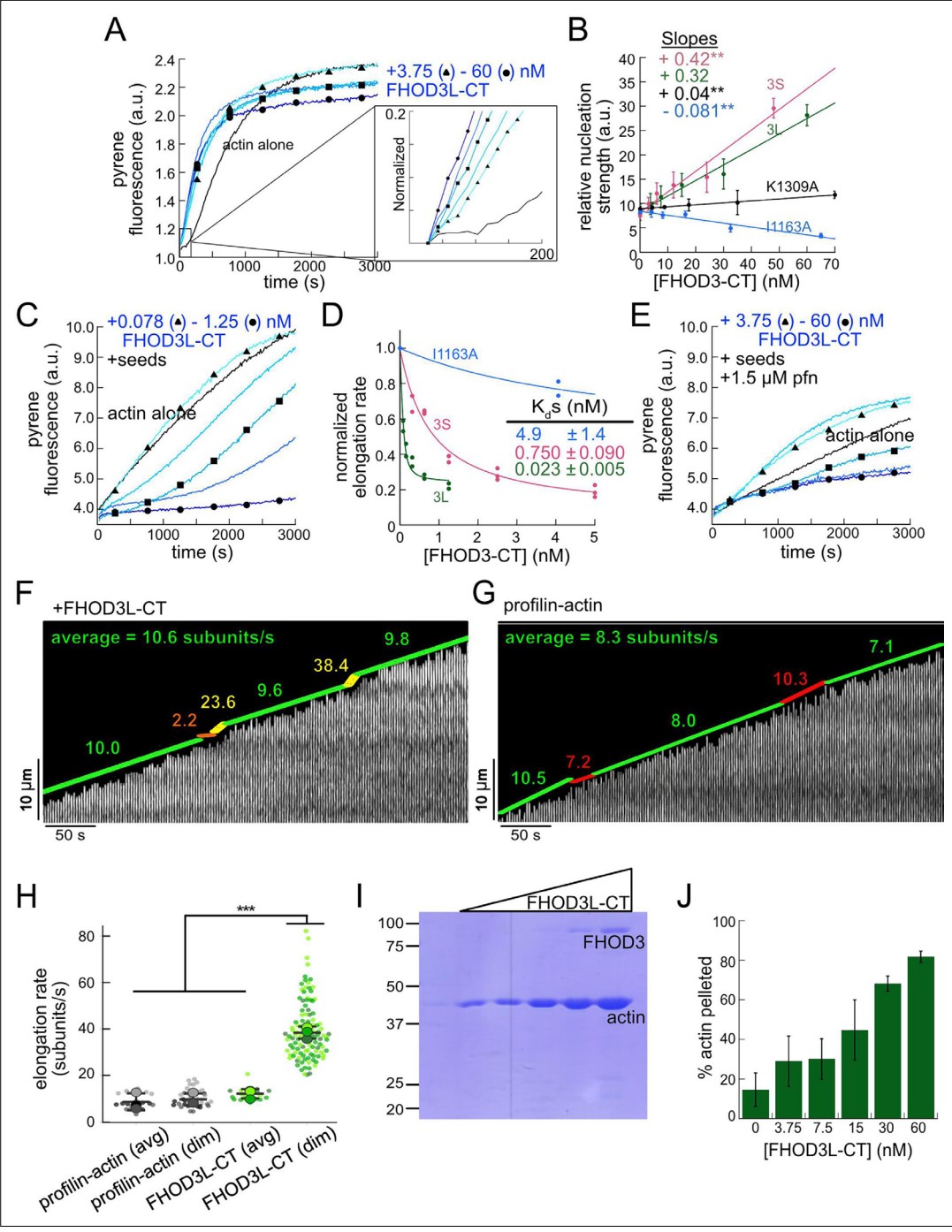

**Figure 2.** Biochemical characterization of human FHOD3L. (**A**) Assembly of 4 µM actin (5% pyrene-labeled) and the indicated concentrations of FHOD3L-CT from a twofold dilution series. (Higher concentrations of FHOD3L-CT are darker shades of blue. Symbols reflect the highest, middle, and lowest concentrations tested. In this case, circle = 60 nM, square = 15 nM, and triangle = 3.75 nM FHOD3LT-CT.) The inset shows the first 200 s of data normalized to the plateau of actin alone. (**B**) Relative nucleation activities of FHOD3L-CT (n=5), FHOD3S-CT (n=4), FHOD3L-CT K1309A (n=3), and I1163A (n=3). Nucleation strength is described by the rate of assembly (i.e. slopes from traces like those in **A**), at an early timepoint, when nucleation dominates ($t_{1/8}$), as a function of formin added. Data points are means, and error bars are standard deviations. Slopes reported are the average slopes of independent experiments. The asterisks indicate significance of difference from FHOD3L-CT (see below for details). (**C**) Barbed-end elongation assay. Final conditions were 0.25 µM F-actin seeds (~0.1 nM barbed ends), 0.5 µM actin (10% pyrene-labeled), and indicated concentrations of FHOD3L-CT. (**D**) Quantification of barbed-end

*Figure 2 continued on next page*

*Figure 2 continued*

affinity for FHOD3L-CT (from **C**), FHOD3S-CT (from *Figure 2—figure supplement 2B*), and FHOD3L-CT I1163A (see *Figure 2—figure supplement 1C* for extended axes). Raw data are shown, and lines are fit to all data points. The K$_d$s reported are the averages of three independent trials (n=3, each; mean ± SD). (**E**) Barbed-end elongation assay with profilin. Final conditions as in (**C**) plus 1.5 µM *Schizosaccharomyces pombe* profilin. (**F**) Kymograph of a growing filament from a total internal reflection fluorescence (TIRF) assay with FHOD3L-CT. Conditions: 1 µM actin (10%-Alexa Fluor 488-labeled), 5 µM Hs profilin-1, and 0.1 nM FHOD3L-CT. The green lines indicate bright regions of growing filament. The yellow lines represent dim regions. The orange line is a pause. Rates calculated for each region are reported as subunits/s. (**G**) Kymograph of a growing filament from a TIRF assay without added FHOD3L-CT. Conditions: 1 µM actin (10%-Alexa Fluor 488-labeled), 5 µM Hs profilin-1. The green lines indicate bright regions of growing filament. The red lines represent dim regions. Rates calculated for each region are reported as subunits/s. (**H**) Elongation rates from TIRF assays. Average elongation rates (10–100 s of seconds) and elongation rates from brief (1–10 s) dim regions are shown separately (n=21, profilin-actin [avg]; n=27 profilin-actin [dim]; n=20, FHOD3L-CT [avg]; n=112, FHOD3L-CT [dim]; 3 channels for all samples; mean ± SD). (**I**) Coomassie-stained polyacrylamide gel of pellet fractions from low-speed bundling assays with 5 µM actin and 0–60 nM FHOD3L-CT. (**J**) Quantification of bundling from (**I**) via densitometry (n=3 each group; mean ± SD). *p<0.05, **p<0.001, ***p<0.0001. p-Values were determined by one-way ANOVA with post hoc Tukey test.

The online version of this article includes the following video, source data, and figure supplement(s) for figure 2:

**Source data 1.** PDF file containing original file for Coomasie stained SDS-PAGE gels displayed in *Figure 2I*, *Figure 2—figure supplement 2H*, indicating the relevant bands and treatments.

**Source data 2.** Original file for Coomasie-stained SDS-PAGE gels displayed in *Figure 2I*, *Figure 2—figure supplement 2H*.

**Figure supplement 1.** Extended biochemical characterization of human FHOD3L.

**Figure supplement 2.** Comparison of FHOD3S to FHOD3L.

**Figure 2—video 1.** FHOD3S/L-CT pause actin filament elongation before brief acceleration.
https://elifesciences.org/articles/104048/figures#fig2video1

---

preceded the burst, lasting ~12 s on average (*Figure 2—figure supplement 1F*). To confirm that the short dim regions were not simply fluorescence inhomogeneities, which are also observed in these assays, we analyzed filaments growing in the absence of formin. Here, the velocity of dim regions was indistinguishable from the average velocity of the filaments, and no pauses were observed (*Figure 2G and H*). In addition, the frequency of dim spots was substantially lower (*Figure 2—figure supplement*

**Table 1.** Summary of biochemical measurements for FHOD3L-CT and mutants.

All data shown are means ± standard deviation, each from at least three independent experiments. n.d.=no data. Light gray columns are data acquired with bulk pyrene-actin-based assays without profilin added. Blue columns provide data from total internal reflection fluorescence (TIRF) analysis of individual filaments (profilin was present). The purple column reports data from the co-sedimentation assay.

| | Pyrene-actin-based assays | | TIRF-based assays | | | Co-sedimentation |
|---|---|---|---|---|---|---|
| | Nucleation strength (a.u./s) | Barbed-end binding affinity (nM) | Elongation rate (subunits/s) | Run length (µm) | Capping duration (s) | Bundling (% actin pelleted at 60 nM) |
| FHOD3L-CT | 0.32 ± 0.04 | 0.028 ± 0.005 | 39 ± 13 | 1.10 ± 0.49 | 11.8 ± 6.8 | 81.7 ± 2.9 |
| FHOD3S-CT | 0.42 ± 0.04* | 0.750 ± 0.090* | 33 ± 12 | 0.90 ± 0.33* | 12.0 ± 7.8 | 56 ± 17 |
| FHOD3L-CT I1163A | –0.081 ± 0.002* | 4.9 ± 1.4* | n.d. | n.d. | n.d. | n.d. |
| FHOD3L-CT K1309A | 0.04 ± 0.02* | n.d. | n.d. | n.d. | n.d. | n.d. |
| FHOD3L-CT K1193L | 0.10 ± 0.02* | 0.470 ± 0.090* | 38 ± 12 | 1.11 ± 0.42 | 12.7 ± 5.6 | 30.8 ± 0.7 |
| FHOD3L-CT GS-FH1 | Nucleates similarly via TIRF nucleation assay | 0.218 ± 0.016* | n.d. | n.d. | n.d. | 61.7 ± 0.7 |

*Statistically different from FHOD3L-CT. Analysis by ANOVA and Tukey post hoc tests, p<0.05. More details are in the figure legends.

*1G*). FHOD3S-CT induced bursts similar to FHOD3L-CT but at slightly lower frequency (*Figure 2—figure supplement 2C–E*). The bursts were slightly, albeit statistically significantly, shorter (characteristic run length of 0.90 ±0.33 µm), while the pauses were indistinguishable from those generated by FHOD3L-CT (*Figure 2—figure supplement 2F and G*). We note that the resolution of our data is at the limit for describing these events. It follows that the metrics must be taken as our best estimates. Additional experiments are needed for more rigorous measurements of the elongation rate and characteristic run length. Furthermore, we cannot exclude that capping, although frequent (preceding 38 of 40 bursts for FHOD3L-CT), could reflect nonspecific interaction with the slide surface. Therefore, we conservatively conclude that, for brief intervals, FHOD3S/L-CT accelerates profilin-actin elongation ~3-fold, following what may be pauses in growth.

Next, we investigated bundling by FHOD3L-CT. To do so, we visually examined the organization of phalloidin-stabilized actin filaments after incubating with FHOD3L-CT. Bundles were readily apparent (*Figure 2—figure supplement 1H*). To better characterize bundling activity, we performed low-speed bundling assays with a fixed concentration of actin filaments and several concentrations of FHOD3L-CT (*Figure 2H and I*). We also compared this activity to bundling by FHOD3S-CT. We found that FHOD3L-CT bundles filaments somewhat more potently than FHOD3S-CT, which is most apparent at concentrations above 15 nM (*Figure 2—figure supplement 2H and I*).

Overall, FHOD3S/L-CT nucleates and elongates actin filaments. FHOD3S-CT is a more potent nucleator, a weaker barbed-end capper, and a slightly weaker bundler compared to FHOD3L-CT (*Table 1*). These data suggest that the acidic T(D/E)$_5$XE insertion of FHOD3L interacts with different surfaces of the actin monomer depending on the biochemical activity. Perhaps the marked increase in capping by FHOD3L-CT is mediated by binding to a basic region exposed at the barbed end of filaments. Decreased activity levels of FHOD3L-CT, such as nucleation and bundling, may be due to the fact that most of the actin surface is acidic.

## Biochemical validation of function-separating mutants

We next designed FHOD3L-CT mutants to separate nucleation and elongation. We also assessed the impact of these mutations on capping and bundling. Previously, Baker et al. identified mutations in the FH2 domain of Bni1 that diminish nucleation while maintaining elongation activity (*Baker et al., 2015*). Based on sequence and structural alignments, FHOD3L K1193 approximates one of these, Bni1 K1467. FHOD3L-CT K1193L nucleated with less than 33% of the strength seen for wild-type (*Figure 3A*). The barbed-end affinity of FHOD3L-CT K1193L was 0.470 ±0.090 nM, ~12-fold weaker than wild-type (*Figure 3B*). Importantly, K1193L-mediated elongation of actin filaments was indistinguishable from wild-type (bursts were again evident, with no significant difference in elongation rate, run length, or capping duration) (*Figure 3C*, *Figure 3—figure supplement 1*). Low-speed bundling assays showed that K1193L bundling is ~2-fold weaker than wild-type (*Figure 3D*). Thus, elongation is maintained while nucleation and bundling are strongly reduced by the K1193L mutation.

To remove FH1-mediated acceleration of elongation, we substituted glycine-serine linkers for the polyproline tracts in FHOD3L-CT (*Zweifel and Courtemanche, 2020*). (The FH2 domain alone was unstable in vitro.) Pyrene-based actin assembly assays were confounded by bundling/aggregation induced by this construct. Although low-speed co-sedimentation assays indicated that the bundling strength of FHOD3L-CT GS-FH1 is slightly weaker than that of wild-type, images show irregular shapes that may scatter light more, thereby disrupting the pyrene signal (*Figure 3D*, *Figure 3—figure supplement 2A*). Therefore, to compare the nucleation strength of FHOD3L-CT GS-FH1 with wild-type, we counted filaments using TIRF microscopy. We observed similar numbers of filaments generated in the presence of low concentrations (used to minimize bundling) of FHOD3L-CT wild-type or GS-FH1 (*Figure 3E*, *Figure 3—figure supplement 2B*). Barbed-end elongation assays in the absence of profilin demonstrated a weakened affinity of FHOD3L-CT GS-FH1 ($K_{app}$ = 0.218 ±0.016 nM) (*Figure 3B*). This finding suggests that the stiffness of the FH1 domain impacts barbed-end binding, but we cannot say whether the effect is direct or indirect. Interestingly, profilin did not decrease barbed-end binding of FHOD3L-CT GS-FH1 ($K_{app}$ = 0.21 ±0.13 nM; *Figure 3F*). Finally, and most importantly, only deceleration was observed in assays with profilin, confirming that elongation was not enhanced by FHOD3L-CT GS-FH1, at any concentration tested (*Figure 3F*). Thus, nucleation is maintained while acceleration of elongation is no longer detected.

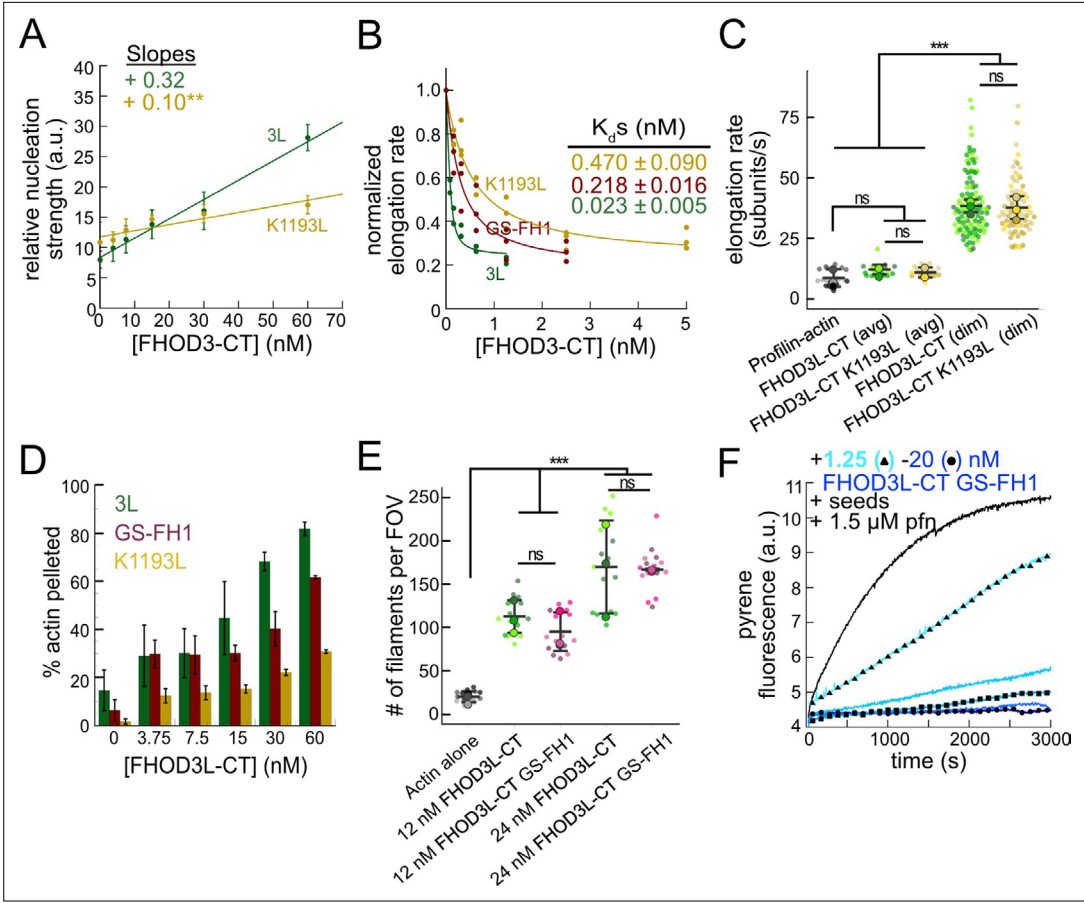

**Figure 3.** Biochemical validation of function-separating mutants. (**A**) Relative nucleation activity for FHOD3L-CT (n=5) and FHOD3L-CT K1193L (n=5). Data points are means, and error bars are standard deviations. Slopes reported are the average slopes of independent experiments. (**B**) Barbed-end affinity measurements for FHOD3L-CT, FHOD3L-CT GS-FH1, and FHOD3L-CT K1193L. Raw data are shown, and the line is a fit to all data points. The $K_d$s reported are the average of three independent trials (n=3, each; mean ± SD). (**C**) Elongation rates from total internal reflection fluorescence (TIRF) assays. Conditions: 1 µM actin (10%-Alexa Fluor 488-labeled), 5 µM Hs profilin-1 ± 0.1 nM FHOD3L-CT or 1 nM FHOD3L-CT K1193L. Average elongation rates (over 10–100 s of seconds) and formin-mediated elongation rates (dim) are shown separately (n=19, FHOD3L-CT K1193L [avg]; n=67, FHOD3L-CT K1193L [dim]; 3 flow channels; mean ± SD, p-values by one-way ANOVA with post hoc Tukey test). (**D**) Quantification of bundling by FHOD3L-CT GS-FH1 and K1193L (n=3, each; mean ± SD). (**E**) Nucleation test. Quantification of the number of filaments per field of view (FOV) for FHOD3L-CT and FHOD3L-CT GS-FH1. Conditions: 4 µM actin with indicated construct. Reaction was diluted in Alexa Fluor 488 Phalloidin to 5 nM actin for visualization. Five images were taken per independent experiment (n=15 images, each; 3 biological replicates, each; mean ± SD, p-values by one-way ANOVA with post hoc Tukey test). Representative images are shown in *Figure 3—figure supplement 2B*. (**F**) Barbed-end elongation assay for FHOD3L-CT GS-FH1 in the presence of profilin. (Higher concentrations of FHOD3L-CT are darker shades of blue. Symbols highlight the highest, middle, and lowest concentrations tested. In this case, circle = 20 nM, square = 5 nM, and triangle = 1.25 nM FHOD3LT-CT.) Final conditions: 0.25 µM F-actin seeds (~0.1 nM barbed ends), 0.5 µM actin (10% pyrene-labeled), 1.5 µM *S. pombe* profilin, and 1.25 nM-20 nM FHOD3L-CT GS-FH1 (n=2; mean ± SD). **p<0.001, ***p<0.0001.

The online version of this article includes the following figure supplement(s) for figure 3:

**Figure supplement 1.** FHOD3L-CT K1193L elongation data.

**Figure supplement 2.** Visualization of nucleation.

Together, these mutations provide a range of nucleation, elongation, capping, and bundling activities (*Table 1*). We, therefore, used them to assess the correlation of biochemical activities with sarcomere formation and function in cardiomyocytes.

## FHOD3L rescues sarcomere organization and contractility in NRVMs

In order to perform structure-function analysis in a cellular context, we established methods to knock down and rescue FHOD3L in NRVMs (similar to *Taniguchi et al., 2009*; *Figure 4A*). Briefly, freshly isolated NRVMs were treated with small interfering RNA (siRNA) targeting FHOD3 using reverse transfection (see Materials and methods for details). Adenoviral infection, driving expression of rescue constructs, was initiated 2 days after siRNA treatment, and cells were examined (fixed or live) 2 days after infection (*Figure 4A*). The rescue constructs contained human FHOD3L, which is not targeted by the siRNA used to remove endogenous rat FHOD3. In all cases, cells were treated with siRNA and adenovirus. For negative controls, AllStars Negative Control siRNA and/or empty virus were substituted for the FHOD3-specific reagents. Thus, we refer to the negative control experiment as a mock knockdown and the knockdown alone as a mock rescue. During assay development, we found that knockdown reduced FHOD3 mRNA to ~35% of original levels after 4 days, with either of two commercially available oligos (*Figure 4—figure supplement 1A*). At this time point, we detected an ~80% reduction in endogenous FHOD3 protein via western blot (*Figure 4B and C*).

We analyzed sarcomere structure in fixed samples. To detect sarcomeres, we stained cells with anti-α-actinin antibodies (*Figure 4D*). To analyze on a per cell basis, we stained plasma membranes with wheat germ agglutinin (WGA) and segmented the NRVMs with CellPose (*Stringer et al., 2021*; *Figure 4—figure supplement 1B*). The DsRed reporter did not provide an accurate readout of exogenous 3xHA-FHOD3L (hereafter, FHOD3L) expression levels (*Figure 4—figure supplement 1C*). Therefore, we used anti-HA antibodies to quantify the expression of exogenous FHOD3 on a per cell basis. By western analysis, we found that average exogenous FHOD3L expression could be as high as ~190% above endogenous levels at the end of the rescue timeline (*Figure 4B and C*). We, therefore, examined the impact of FHOD3L expression level on sarcomere structure. We detected essentially no correlation ($R^2$ ranges from 0.001 to 0.04) of sarcomere number, length, or width over a tenfold change in FHOD3L expression levels (*Figure 4—figure supplement 1D–F*). Despite the evidence that sarcomere structure was not a function of FHOD3 expression levels, we decided to exclude NRVMs that express very highly above endogenous FHOD3 levels. We set an upper cutoff for the normalized exogenous FHOD3L expression per cell area of 5% and applied that same intensity level as a cutoff for all other experiments (*Figure 4—figure supplement 1D*). We also set a lower cutoff slightly above background HA levels to exclude NRVMs not expressing exogenous FHOD3L.

We confirmed that sarcomeres remained organized upon mock knockdown, counting 12 ± 13 sarcomeres per cell (*Figure 4D and E*, *Table 2*). (Measurements from NRVMs are summarized in *Table 2*.) Sarcomeres were largely absent in the mock rescue (3 ± 7) with α-actinin puncta and aggregates visible by immunofluorescence (IF) (*Figure 4D and E*). Expression of wild-type FHOD3L was sufficient to rescue the loss of sarcomeres, and anti-HA staining demonstrated that exogenous FHOD3L localized to sarcomeres, as expected (*Figure 4D*). In fact, FHOD3L-rescued NRVMs formed significantly more sarcomeres than the mock knockdown NRVMs (19 ±14 vs 12 ± 13; *Figure 4E*). We note that the sarcomere number measurement has a high standard deviation because some cells within the expression level cutoffs, including at relatively high levels of detected FHOD3L, lacked sarcomeres (*Figure 4—figure supplement 1F*). Such right-tailed distributions are consistent with other reports of sarcomere numbers per cardiomyocyte (*Neininger-Castro et al., 2023*).

To analyze sarcomere integrity more closely, we measured sarcomere lengths (Z-line to Z-line) and widths (Z-line lengths). In the mock rescue cells, the sarcomeres that remained were both shorter and narrower than those in the mock knockdown cells (*Figure 4F and G*). When rescued with FHOD3L, sarcomere lengths (1.72 ±0.18 μm) and sarcomere widths (1.70 ±0.46 μm) recovered to lengths comparable to the mock knockdown control (*Figure 4F and G*). To measure thin filament length, we stained NRVMs with phalloidin and anti-HA. We observed shorter thin filament lengths in FHOD3L-rescued NRVMs (739 ±81 nm) compared to mock knockdown NRVMs (925 ±94 nm) (*Figure 4H and I*). The difference in thin filament length could reflect the increased number of sarcomeres in the rescued cells and/or could indicate that the sarcomeres have not reached their final steady state after 2 days of FHOD3L expression. Overall, FHOD3L-rescued NRVMs form sarcomeres de novo that well

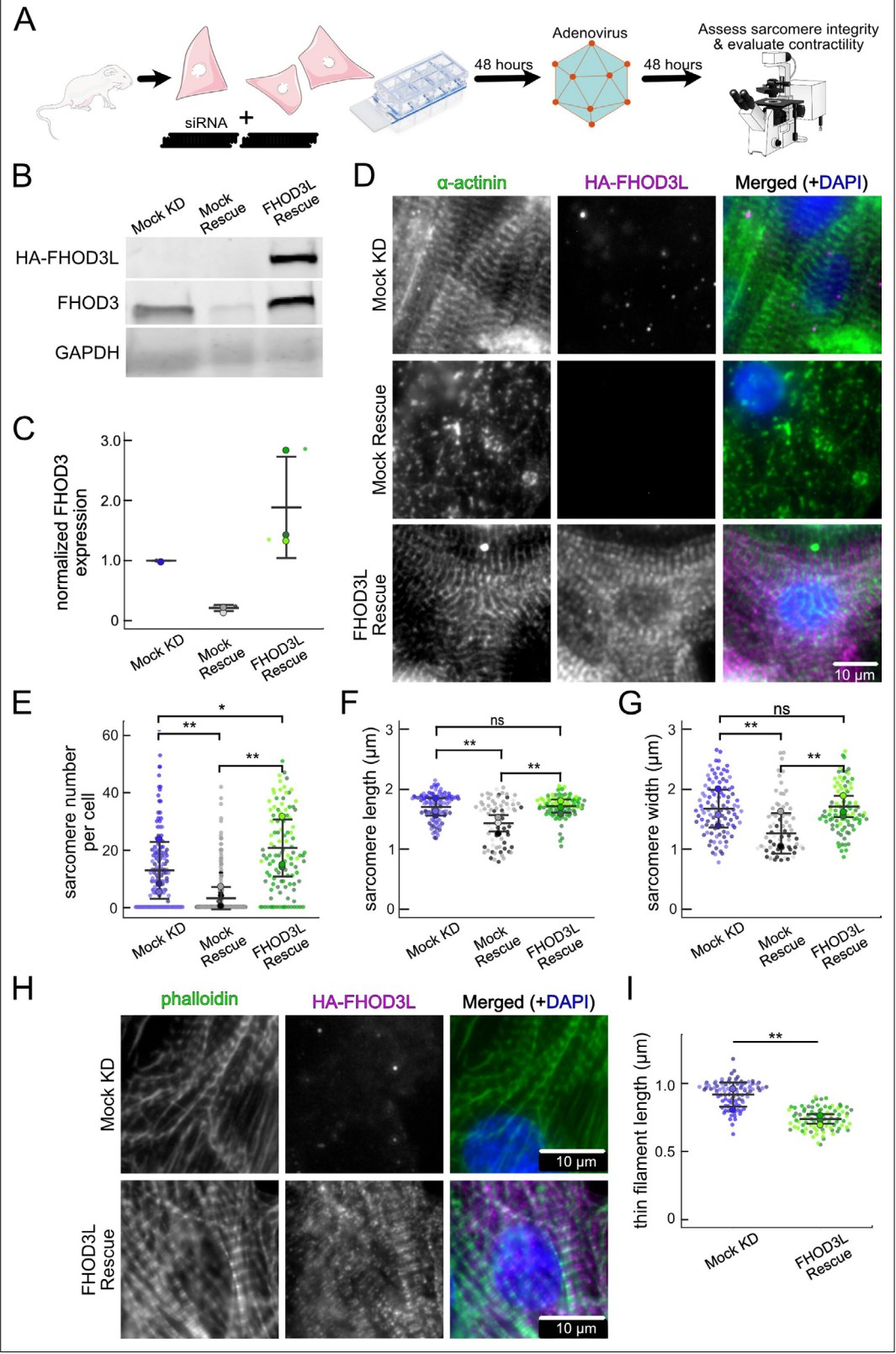

**Figure 4.** FHOD3L rescues sarcomere organization in neonatal rat ventricular myocytes (NRVMs). (**A**) Overview of the rescue-experiment protocol. Reverse transfection of small interfering RNA (siRNA) upon plating NRVMs followed by infection with adenovirus to drive exogenous expression. (**B**) Western blot showing depletion of endogenous FHOD3 after knockdown, and exogenous FHOD3L expression levels after rescue. GAPDH used

*Figure 4 continued on next page*

*Figure 4 continued*

as a loading control. (**C**) Quantification of western blot in (**B**) normalized to GAPDH for each lane and then normalized to endogenous levels in the mock knockdown (n=3, each; mean ± SD). (**D**) Sarcomere integrity indicated by immunofluorescent staining of α-actinin (green). Localization of exogenous HA-FHOD3L is shown in magenta. DAPI (blue) is included in the merged images. Wheat germ agglutinin (WGA) is not shown for clarity. (**E**) Quantification of sarcomere number per NRVM (n=160 cells, mock KD; n=334 cells, mock rescue; n=110 cells, FHOD3L rescue). (**F**) Average sarcomere lengths per NRVM (n=100 cells, mock KD; n=71 cells, mock rescue; n=92 cells, FHOD3L rescue). (**G**) Average sarcomere widths (Z-line lengths) per NRVM (n=100 cells, mock KD; n=71 cells, mock rescue; n=92 cells, FHOD3L rescue). (**H**) Epifluorescent micrographs showing mock knockdown and FHOD3L-rescued NRVMs stained, with phalloidin (green) to visualize thin filaments and anti-HA (magenta) to show expression of exogenous FHOD3L. (**I**) Quantification of thin filament lengths for mock knockdown and FHOD3L-rescued NRVMs (n=94 cells, mock KD; n=84 cells, wild-type [WT] rescue). For (**E, F, G, I**), data from three biological replicates for each condition are represented by different shades. Mean ± SD is shown. p-Values were calculated with Mann-Whitney U tests, *p<0.05; **p<0.001.

The online version of this article includes the following source data and figure supplement(s) for figure 4:

**Source data 1.** PDF files containing original file western blots displayed in *Figure 4B*, indicating the relevant bands and treatments.

**Source data 2.** Originals file for westerns blots displayed in *Figure 4B*.

**Figure supplement 1.** Method establishment for FHOD3L rescues in neonatal rat ventricular myocytes (NRVMs).

approximate the sarcomeres in the negative control. We compared further experimental results to FHOD3L rescue cells, due to the slight differences.

We next measured contractile function in FHOD3L-rescued NRVMs. To do so, we used a motionGUI MATLAB program that makes use of digital image correlation (DIC) to measure contractility (*Huebsch et al., 2015*; *Nakano et al., 2017*). Contraction and relaxation velocities inform us about systolic and diastolic function of the cardiomyocytes, respectively, and thus their ability to promote proper blood flow in organisms (*Ferreira-Martins and Leite-Moreira, 2010*). As expected, maximal contraction and relaxation velocities were both significantly decreased upon FHOD3 knockdown (*Figure 5A,*

**Table 2.** Summary of FHOD3L and mutant rescue experiments in neonatal rat ventricular myocytes (NRVMs).

All data shown are means ± standard deviation, each from three independent experiments (except for GS-FH1 rescue: maximal contraction and relaxation velocity measurements and average % rhythmic contractions from two independent experiments due to one biological replicate not contracting). n.d.=no data. In the lightest columns, '–' indicates treatment with negative control small interfering RNA (siRNA) or empty virus, '+' indicates treatment with FHOD3 siRNA or corresponding FHOD3L adenovirus. Data in the orange columns were acquired from fixed and stained cells. Data in the red columns were acquired from live cells. Statistical analyses are described in the figure legends. 'In the lightest columns' should be 'In the yellow columns'. Data in the 'medium-colored columns' should be 'Data in the orange columns'. 'Data in the darkest columns' should be 'Data in the red columns'.

| | Treatment | | Fixed samples | | | | Live samples | | | |
|---|---|---|---|---|---|---|---|---|---|---|
| | FHOD3 KD? | FHOD3 AdV? | Avg sarcomere number per NRVM | Sarcomere length (µm) | Z-line length (µm) | Thin filament length (nm) | Maximal contraction velocity (µm/s) | Maximal relaxation velocity (µm/s) | Avg % rhythmic contractions | Avg % contracting NRCs in FOV |
| Mock knockdown | – | – | 12 ± 13 | 1.71 ± 0.22 | 1.69 ± 0.43 | 925 ± 94 | 7.9 ± 1.6 | 6.0 ± 1.7 | 74 ± 9 | 98.2 ± 3.2 |
| Mock rescue | + | – | 3 ± 7* | 1.46 ± 0.37* | 1.38 ± 0.46* | n.d. | 6.0 ± 1.7* | 4.4 ± 1.5* | 14 ± 12 | 89 ± 11 |
| FHOD3L rescue | + | + | 19 ±14*, † | 1.72 ± 0.18† | 1.71 ± 0.46† | 739 ±81*, † | 7.5 ± 1.5† | 6.1 ± 1.5† | 72 ± 20 | 97.2 ± 4.8 |
| K1193L rescue | + | + | 17 ± 17 | 1.69 ± 0.16 | 1.73 ± 0.41 | 792 ± 79 ‡ | 7.7 ± 1.6 | 5.9 ± 1.6 | 88 ± 21 | 98.2 ± 3.2 |
| GS-FH1 rescue | + | + | 2 ± 4 ‡ | 1.42 ± 0.35 ‡ | 1.23 ± 0.29‡ | n.d. | 5.6 ± 1.7‡ | 3.6 ± 1.2 ‡ | 3.7 ± 5.2 | 32 ± 32 |
| FHOD3L overexpression | – | + | n.d. | n.d. | n.d. | n.d. | 6.6 ± 1.6‡ | 5.5 ± 1.6 ‡ | 72 ± 21 | 76 ± 32 |

*Statistically different from mock KD (mock rescue and Fhod3L rescue).

†Statistically different from mock rescue (FHOD3L and GS-FH1).

‡Statistically different from FHOD3L rescue (all mutant rescues and overexpression).

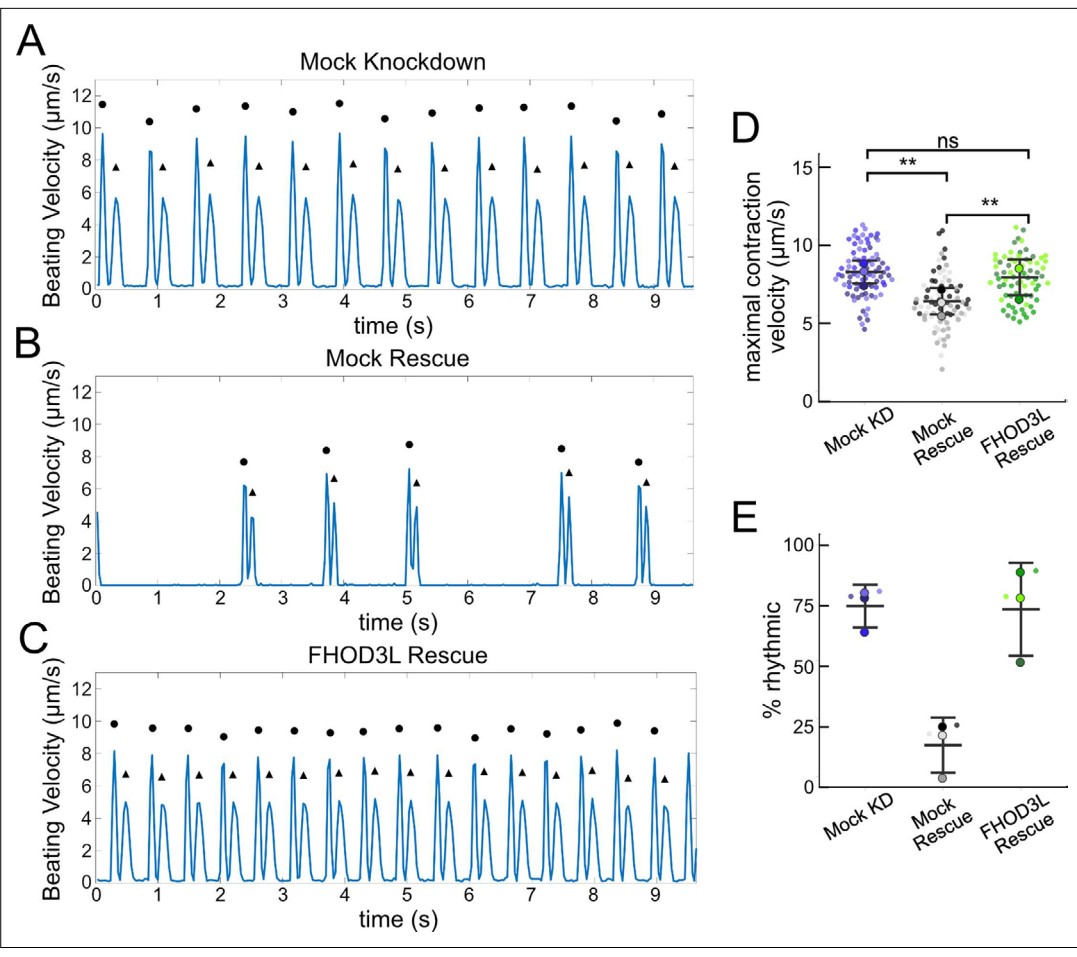

**Figure 5.** FHOD3L rescues contractility in neonatal rat ventricular myocytes. (**A–C**) Motion analysis by digital image correlation. Mock knockdown, mock rescue, and FHOD3L rescue beating patterns are shown. The first (•) and the second (▲) peak of each duplex represent the contraction and the relaxation, respectively. (**D**) Maximal contraction velocities quantified for mock knockdown, mock rescue, and FHOD3L wild-type rescue (n=82 regions of interest [ROIs], mock KD, n=81 ROIs, mock rescue, n=72, ROIs, FHOD3L rescue; 3 biological replicates, each; mean ± SD, p-values by Student's two-sample, unpaired t-test). (**E**) Quantification of the percentage of analyzed ROIs from the videos that contained rhythmic contractions for mock knockdown, mock rescue, and FHOD3L wild-type rescue (n=3, each; mean ± SD). **p<0.001.

The online version of this article includes the following video and figure supplement(s) for figure 5:

**Figure supplement 1.** Relaxation velocity and fraction contracting.

**Figure supplement 2.** Overexpression of FHOD3L.

**Figure 5—video 1.** Exogenous FHOD3L expression rescues rhythmic contractions in neonatal rat ventricular myocytes (NRVMs).

https://elifesciences.org/articles/104048/figures#fig5video1

---

*B, and D*, *Figure 5—figure supplement 1A*). Both velocities recovered to mock knockdown levels by subsequent expression of FHOD3L (*Figure 5A, C, and D*, *Figure 5—figure supplement 1A*, *Figure 5—video 1*). We also examined the expression of FHOD3L in an otherwise untreated background (overexpression). Intensity of exogenous FHOD3L per NRVM was notably lower compared to the wild-type rescue despite infecting with the same titer, suggesting that NRVMs attempt to maintain FHOD3L levels below some maximum (*Figure 5—figure supplement 2A*). Consistent with this idea, contraction and relaxation velocities were reduced upon overexpression of FHOD3L (*Figure 5—figure supplement 2B and C*). These data further demonstrate that the expression levels in our rescue experiments, while higher on average than endogenous levels, are within an acceptable range.

To assess the impact of FHOD3L depletion on cardiac rhythm, we measured the percentage of beating area in each video that exhibited consistent, rhythmic contractions (defined as no more than 1 beat out of sync in a 10 s period). We observed rhythmic contractions for both the mock knock-down NRVMs and FHOD3L-rescued NRVMs for ~75% of the analyzed videos (*Figure 5A, C, and E*, *Figure 5—video 1*). In contrast, primarily arrhythmic contractions were detected in FHOD3-depleted NRVMs, with only ~15% of the videos showing rhythmic beating (*Figure 5B and E*, *Figure 5—video 1*). To assess whether some NRVMs in these conditions were not contracting at all, we estimated the proportion of NRVMs that were contracting per video and found contractions throughout ~95% of the field of view on average for both mock knockdown NRVMs and wild-type FHOD3L-rescued NRVMs (*Figure 5—figure supplement 1B*). For FHOD3-depleted NRVMs, we observed a reduction in contractile area to ~80% (*Figure 5—figure supplement 1B*). This value was higher than expected based on other metrics (*Figure 5D and E*). We attribute some of the movement to neighboring cells pulling each other. We also quantified contractility when we overexpressed FHOD3L. FHOD3L overexpression did not clearly impact the rhythmic contractions or the area of contraction, though variance increased (*Figure 5—figure supplement 2D and E*), consistent with the idea that, at high enough levels, FHOD3L activity can be detrimental in NRVMs.

## Loss of nucleation, but not elongation, is tolerated for sarcomere formation and cardiac function

Once we established baseline levels of sarcomere structure and contractility in NRVMs with the wild-type FHOD3L rescue, we asked which actin organizing activities of FHOD3L are important for its cellular function. To this end, we performed rescue experiments with the nucleation-hindering mutant (K1193L) and the elongation-hindering mutant (GS-FH1). Expression of the K1193L mutant resulted in expression levels similar to wild-type (*Figure 6—figure supplement 1A and B*). Little to no correlation between FHOD3 expression level and sarcomere metrics was observed ($R^2$ ranges from 0.003 to 0.2; *Figure 6—figure supplement 1D*). Overall, cells expressing FHOD3L K1193L were almost indistinguishable from those expressing FHOD3L. FHOD3L K1193L localization was striated (*Figure 6A*), and the number of sarcomeres per NRVM was rescued to wild-type levels (15 ± 16) (*Figure 6B*). Sarcomere lengths and widths were indistinguishable from wild-type (*Figure 6C and D*). The only statistically significant difference was thin filament length, which was ~7% longer in the K1193L-rescued NRVMs (792 ±79 nm) compared to those in FHOD3L-rescued NRVMs (739 ± 81) (*Figure 6A' and E*, *Figure 6—figure supplement 2A*). In agreement with the well-organized, near wild-type appearance of sarcomeres, contraction and relaxation velocities, as well as the proportion of rhythmically contracting NRVMs, were indistinguishable from the FHOD3L rescue (*Figure 6F and G*, *Figure 6—figure supplement 2B and C*, *Figure 6—video 1*). Thus, ~70% loss of nucleation, ~20-fold weaker barbed-end capping, and loss of over 50% of bundling activity had no deleterious impact on FHOD3L's ability to function in NRVMs.

In contrast, the elongation-deficient mutant (GS-FH1) did not rescue FHOD3L loss in NRVMs. Based on total fluorescence per cell, we determined that GS-FH1 levels were lower than observed for wild-type or K1193L, despite infecting a similar proportion of NRVMs (*Figure 6—figure supplement 1A*). In most cells, the FHOD3L GS-FH1 localization was diffuse (*Figure 6A*, *Figure 6—figure supplement 2B*). In a few cells that probably escaped from RNAi knockdown, we observed striated doublets of FHOD3L GS-FH1 between Z-lines, demonstrating that the protein can fold and localize correctly if sarcomeres are intact (*Figure 6—figure supplement 2C*). The turnover rate of GS-FH1 may be higher than wild-type intrinsically or because there were few binding sites available, i.e., fewer sarcomeres to which GS-FH1 could bind. Phalloidin staining revealed actin puncta in most cells expressing GS-FH1 (*Figure 6A'*, *Figure 6—figure supplement 2C*). The puncta were smaller than expected for mature sarcomeres but were aligned in some cases, suggesting the possible formation of premyofibrils.

Importantly, GS-FH1 levels detected and analyzed were within the acceptable intensity range we established for FHOD3L (*Figure 6—figure supplement 1A and B*). As was true for the other rescues, there was no correlation between GS-FH1 expression levels and measured sarcomere metrics ($R^2$ ranges from 0.002 to 0.07; *Figure 6—figure supplement 1E*). To further test for any impact due to low protein levels, we identified two subpopulations of FHOD3L rescue cells with HA intensity distributions similar to that of the GS-FH1 cells: the first biological replicate and all cells with HA intensity levels less than 720 a.u./µm² (*Figure 6—figure supplement 3A*). We found no significant correlations

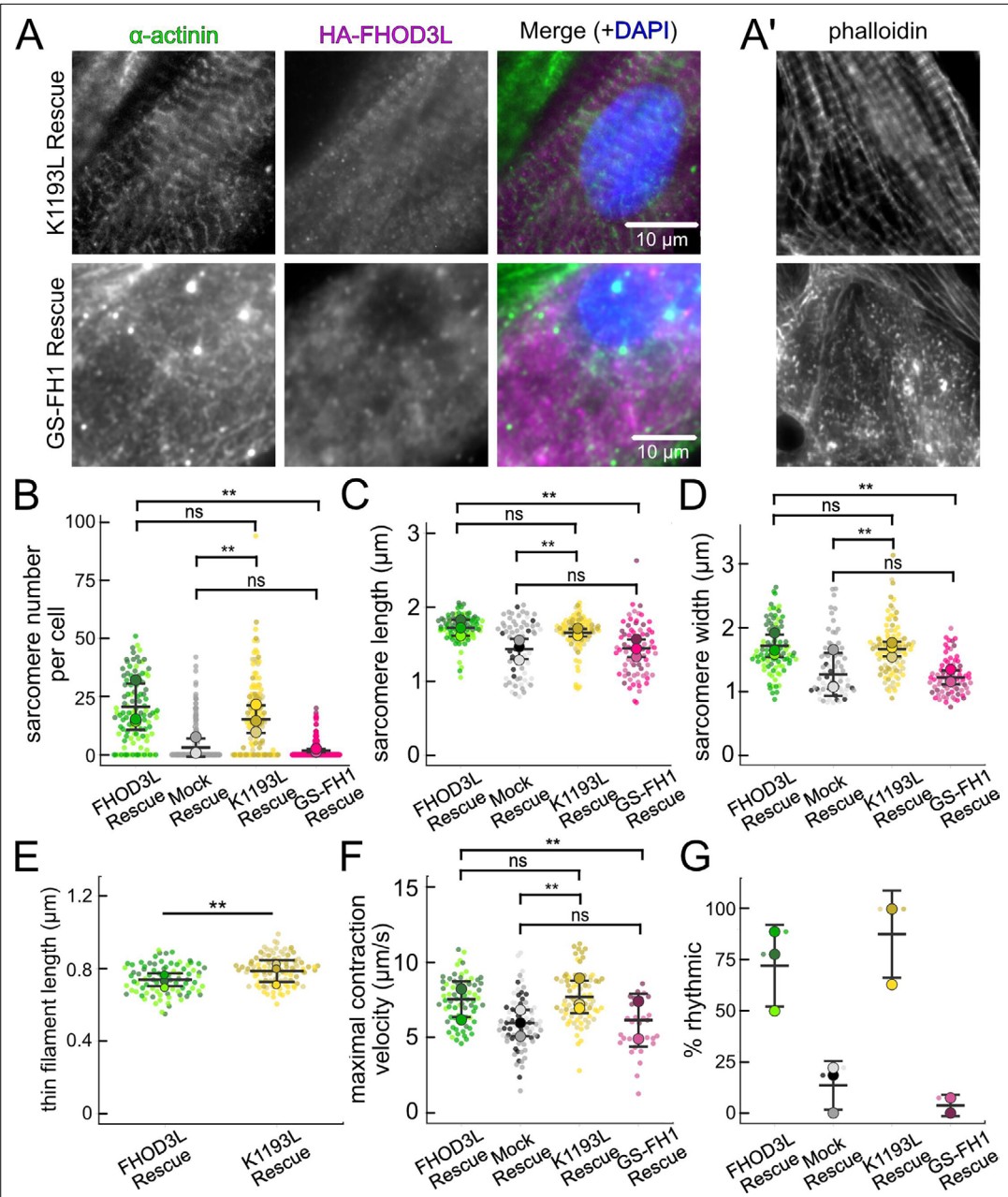

**Figure 6.** Loss of nucleation, but not elongation, is tolerated for sarcomere formation and function. (**A**) Images of neonatal rat ventricular myocytes (NRVMs) rescued with K1193L or GS-FH1. Sarcomere integrity indicated by immunofluorescent staining of α-actinin (green). Localization of exogenous HA-FHOD3L is shown in magenta. DAPI (blue) is included in the merged images. Wheat germ agglutinin (WGA) is not shown for clarity. (**B**) Quantification of sarcomere number per NRVM in the FHOD3L, K1193L, and GS-FH1 rescues (n=148 cells, K1193L; n=259 cells GS-FH1; 3 biological replicates; mean ± SD, p-values by Mann-Whitney U test). (**C**) Average sarcomere lengths per NRVM in the FHOD3L, K1193L, and GS-FH1 rescues (n=95 cells, K1193L; n=73 cells, GS-FH1; 3 biological replicates; mean ± SD, p-value for FHOD3L comparison to GS-FH1 by Student's two-sample, unpaired t-test, all other p-values by Mann-Whitney U test). (**D**) Average sarcomere widths (Z-line lengths) per NRVM in the FHOD3L, K1193L, and GS-FH1 rescues (n=95 cells, K1193L; n=73 cells, GS-FH1; 3 biological replicates; mean ± SD, p-values by Mann-Whitney U test). (**E**) Quantification of thin filament lengths for FHOD3L and K1193L-rescued NRVMs (n=99 cells, K1193L; 3 biological replicates, each; mean ± SD, p-value by Mann-Whitney U test). (**F**) Quantification of contracting NRVMs, as in *Figure 5D*, for FHOD3L, K1193L, and GS-FH1-rescued NRVMs (n=3, each; mean ± SD). (**G**) Quantification of rhythmic contractions, as in *Figure 5E*, for FHOD3L, K1193L, and GS-FH1-rescued NRVMs (n=3, FHOD3L and K1193L; n=2, GS-FH1; mean ± SD). **p<0.001.

*Figure 6 continued on next page*

*Figure 6 continued*

The online version of this article includes the following video and figure supplement(s) for figure 6:

**Figure supplement 1.** FHOD3L expression level does not predict sarcomere metrics.

**Figure supplement 2.** Potential pre-myofibrils in FHOD3L GS-FH1 rescue cells.

**Figure supplement 3.** Expression levels of GS-FH1 do not predict phenotype.

**Figure 6—video 1.** Expression of FHOD3L function separating mutants in neonatal rat ventricular myocytes (NRVMs).

https://elifesciences.org/articles/104048/figures#fig6video1

between HA intensity levels and sarcomere metrics (*Figure 6—figure supplement 3C–E*). Furthermore, we found no difference in sarcomere numbers or Z-line lengths for these subgroups compared to the whole collection of FHOD3L cells (*Figure 6—figure supplement 3B*). There was a statistically significant, albeit small (~6%), decrease in the sarcomere lengths of cells in replicate 1, but sarcomere lengths in the HA<720 group were indistinguishable from the full FHOD3L dataset. Based on these findings, we conclude that the expression levels in the GS-FH1-rescued NRVMs are high enough to rescue the FHOD3 knockdown, in principle, supporting our conclusion that the defect is due to loss of elongation activity.

The sarcomere number per GS-FH1 rescue cell was only 2 ± 4, similar to the mock rescue (*Figure 6A and B*). Within the sarcomeres detected, sarcomere lengths and widths were both reduced (*Figure 6C and D*). We were not able to measure thin filament lengths with confidence, due to the lack of sarcomere organization. Not surprisingly, contractility was nearly abolished after rescuing with GS-FH1. Contraction and relaxation velocities were decreased, and contractions were largely arrhythmic (*Figure 6F and G*, *Figure 6—figure supplement 2D and E*, *Figure 6—video 1*). We note that barbed-end capping and bundling activities were weaker than wild-type in the GS-FH1 mutant, but they were equivalent to or greater than the activities found for the K1193L mutant, which rescued well (*Figure 3B and D*). Therefore, we conclude that the elongation activity of FHOD3L is the primary activity required for sarcomere formation, whereas reduced nucleation, barbed-end binding, and bundling activities are well tolerated.

## Discussion
### FHOD3L elongation is necessary for sarcomeres in NRVMs

Published rates of nucleation and elongation by formins both vary by at least an order of magnitude. It is generally thought that the specific actin assembly properties of a given formin are set for its function, i.e., the structure this formin will build. To directly test this idea, Homa et al. replaced Cdc12p, the formin critical to *S. pombe* cytokinesis, with several chimeras of differing nucleation strength (*Homa et al., 2021*). Indeed, they found a strong positive correlation between formins with nucleation strength similar to Cdc12p and their ability to drive cytokinesis. In a computational model that recapitulates yeast cable structure, it was found that nucleation and/or elongation could be tuned to build the cables (*McInally et al., 2021*). Thus, how formins are used is likely to differ from case to case.

We determined that FHOD3L can nucleate in vitro, albeit weakly compared to many formins. Unlike in cytokinesis, we found that this activity is not necessary for its function in NRVMs. The necessity of actin assembly, nucleation, and/or elongation, by Fhod-family formins, has been questioned but supported by the failure of the IA mutation to rescue in multiple species (*Kan et al., 2012a*; *Kimmich et al., 2024*; *Kutscheidt et al., 2014*; *Shwartz et al., 2016*; *Taniguchi et al., 2009*). However, the IA mutation is thought to diminish both nucleation and elongation. To assess the importance of nucleation more specifically, we tested a mutant variant that substantially reduces the nucleation activity without altering elongation (FHOD3L K1193L). This mutant had almost no impact on sarcomere formation or function in NRVMs. The only statistically significant difference we detected was in thin filament length (*Figure 6E*). A host of proteins is critical to establishing and maintaining thin filaments at a precise length (*Szikora et al., 2022*). No other studies have implicated FHOD3L in this process to date. Thus, we believe that the length difference is more likely to reflect the state of maturity of the cells (time post-infection) and/or the number of sarcomeres per cell. While we only have three data

points at this time, sarcomere number and thin filament length were highly correlated (inversely) in our experiments, with an $R^2$ of 0.90.

By rescuing NRVMs with FHOD3L K1193L, we also tested a formin that caps barbed ends with a $K_d$ that is ~15 times higher than wild-type (i.e. ~15× weaker) and bundles filaments ~50% less potently than wild-type. The fact that the cardiomyocytes were indistinguishable from wild-type when expressing this mutant suggests that neither of these activities is required, and, certainly, they are not needed at wild-type levels. Assuming that FHOD3L activities are tuned to meet its in vivo functions, these magnitudes of change in activity would be expected to result in measurable phenotypes in function and/or morphology of the cell.

The fact that K1193L has no impact on the elongation properties of FHOD3L in vitro and still rescues sarcomeres leads to an indirect conclusion that elongation is the actin assembly activity that is important in NRVMs. Of course, the failure to rescue by FHOD3L GS-FH1, an elongation incompetent mutant, strongly supports this conclusion (*Figure 6*). The elongation properties of FHOD3L are unusual. Its elongation rate is among the highest known, including mDia1 and Cappuccino (*Bor et al., 2012*; *Kovar et al., 2006*). However, processivity is very brief, resulting in filaments of only ~1 µm. We previously found that the *Drosophila* splice variant FhodA has a similarly short characteristic run length (*Patel et al., 2018*). In principle, this characteristic run length is sufficient to build sarcomeric thin filaments, which are about 1 µm long in NRVMs. Notably, a splice variant of *Drosophila* Fhod that only differs by having a shorter tail, FhodB, typically elongates for ~20 µm, and the tails of mammalian and *Drosophila* formins are highly conserved (*Bremer et al., 2024*). These observations lead us to question whether the processivity of Fhod-family formins is somehow regulated in vivo such that it can build short filaments when needed and longer filaments in other contexts. Interestingly, a computational model describing Bni1-built actin cables demonstrates that tuning the length of individual filaments (set by elongation rate and processivity) to cell size is sufficient to account for cable differences in cells of different sizes (*McInally et al., 2024*).

The FH1 domain strongly influences the elongation rate in vitro. In NRVMs, we found that the replacement of the FH1 domain with a flexible linker (FHOD3L GS-FH1) resulted in a severe phenotype. These cells had very few sarcomeres. FHOD3L GS-FH1 was indistinguishable from mock rescue in sarcomere number, structure (shorter and narrower), and function (contraction and relaxation velocities are not different). The result could be because the expression level of FHOD3L GS-FH1 was lower, on average, than those for the other constructs. We argue against this based on multiple observations: First, similar expression levels of wild-type FHOD3L are sufficient to fully rescue the knockdown (*Figure 6—figure supplement 3*). Second, despite similar metrics in sarcomere structure and function, the rhythmicity of contraction was markedly diminished compared to the mock rescue (*Figure 6G*). Third, we do not believe that the low protein levels reflect protein instability. We detect FHOD3L GS-FH1 bound to residual sarcomeres, suggesting that it folds and is stable when binding sites within sarcomeres are available (*Figure 6—figure supplement 2C*). It is possible that the striated appearance of FHOD3L GS-FH1 in cells with sarcomeres is the result of protein stabilization through heterodimerization with residual endogenous FHOD3L. However, we favor a model in which low FHOD3L GS-FH1 levels reflect increased protein turnover due to the absence of sarcomeres.

Recent work in *Caenorhabditis elegans* is consistent with our findings. Kimmich et al. removed the endogenous coding region of the FH1 domain in worm Fhod-1 (the only worm Fhod gene) (*Kimmich et al., 2024*). Body wall muscle is severely impacted by this deletion. In addition, they found that profilin is required for Fhod-1 function. Together, these data strongly argue that elongation activity is important. Interestingly, the muscle phenotype is more severe in a strain predicted to delete part of the FH2 domain and the downstream sequence, but not the FH1 domain, *fhod-1(tm2363)* (*Kimmich et al., 2024*; *Mi-Mi et al., 2012*). Thus, nucleation activity may also be necessary in this animal. Ultimately, complete removal of any domain, as we do here with the GS-FH1 mutant, may be too blunt of an approach. We expect experiments with modified elongation (e.g. slower or more processive) to provide further insight.

## Why does FHOD3L elongate in NRVMs?

Given these new data, what could FHOD3L be doing in the cell? In multiple species, premyofibrils are built, but they fail to mature into wild-type myofibrils in the absence of FHOD (*Kan et al., 2012a*; *Kimmich et al., 2024*; *Shwartz et al., 2016*). It follows that FHOD3 is not required to build the earliest

thin filaments in the sarcomere. Instead, FHOD3 could contribute new thin filaments to expanding, maturing sarcomeres. If this were the case, we might expect nucleation to be more important than our data indicate. Instead, FHOD3L could be responsible for elongating filaments that are nucleated by a different protein. For example, FHOD3L and DAAM could collaborate, or they could contribute to the apparent redundancy and robustness built into sarcomerogenesis (*Szikora et al., 2022*). DAAM knockout mice have overlapping cardiac phenotypes with those of the FHOD3 knockout (*Kan et al., 2012a*; *Li et al., 2011*). Furthermore, a lack of sarcomere thickening and loss of sarcomere organization are common phenotypes in *Drosophila* knockdowns of Daam1 and Fhod (*Molnár et al., 2014*; *Shwartz et al., 2016*). While FHOD3L and DAAM cannot compensate for the loss of one or the other, they might help when one is compromised.

Alternatively, FHOD3L could be reinforcing the Z-line structure to facilitate sarcomere thickening. This idea is consistent with the narrow myofibrils and Z-body disorder observed in worms lacking functional Fhod-1 (*Kimmich et al., 2024*; *Mi-Mi et al., 2012*). If capping and bundling were important to the Z-line integrity, one would expect to find FHOD3L localized in the Z-line. However, FHOD localization varies between species and with development, suggesting that this role is not conserved, if it exists. In mouse heart tissue, FHOD3L appears to bind directly to MyBP-C, which localizes between the Z- and M-lines (*Matsuyama et al., 2018*), suggesting that it is not actively stabilizing the Z-line. Thus, transient actin assembly may be sufficient, consistent with our findings that capping and bundling are not essential in NRVMs.

# Materials and methods

## Key resources table

| Reagent type (species) or resource | Designation | Source or reference | Identifiers | Additional information |
|---|---|---|---|---|
| Strain, strain background (*Escherichia coli*) | BL21(DE3) | Novagen/Sigma-Aldrich | 69,450-M RRID:Ecoli_0001 | |
| Cell line (*Homo sapiens*) | HEK293T | Dr. Kohnosuke Mitani, UCLA | CRL-3216 | |
| Transfected construct and biological sample (human) | pAdenoX-CMV-3xHA-FHOD3L-CMV-DsRed (plasmid and adenovirus) | Takara Bio; this paper | Cat. No. 632262 | Adenoviral construct to transfect and purify the adenovirus to express FHOD3L wild-type in NRVMs. |
| Biological sample (human) | pAV-CMV-{3xHA-FHOD3L GS-FH1}:SV40 pA-CMV-mCherry (adenovirus) | VectorBuilder | Cat#AVS(VB230718-1114xtg) | Adenovirus to transfect and express FHOD3L GS-FH1 in NRVMs. |
| Biological sample (human) | pAV-CMV-{3xHA-FHOD3L K1193L}:SV40 pA-CMV-mCherry (adenovirus) | VectorBuilder | Cat#AVS(VB230528-1145fkg) | Adenovirus to transfect and express FHOD3L K1193L in NRVMs. |
| Sequence-based reagent | siRNA against rat FHOD3 | QIAGEN | Rn_LOC100360334_2 Flexitube siRNA | |
| Sequence-based reagent | AllStars Negative Control siRNA | QIAGEN | Cat. No. 1027281 | |
| Sequence-based reagent | GAPDH_F | This paper | qPCR primers | CCGCATCTTCTTGTGCAGTG |
| Sequence-based reagent | GAPDH_R | This paper | qPCR primers | CGATACGGCCAAATCCGTTC |
| Sequence-based reagent | FHOD3_F | This paper | qPCR primers | CAGCCAATCACGGAG |
| Sequence-based reagent | FHOD3_R | This paper | qPCR primers | TGCTGTCCTTGCCCTGA |
| Biological sample (*Rattus norvegicus*) | Primary neonatal rat ventricular myocytes | UCLA Cardiovascular Research Theme Core | | Freshly isolated from male and female Sprague-Dawley rats |

*Continued on next page*

*Continued*

| Reagent type (species) or resource | Designation | Source or reference | Identifiers | Additional information |
|---|---|---|---|---|
| Antibody | anti-α-actinin (Mouse monoclonal) | Sigma | Cat. No. A7811, RRID:AB_476766 | IF (1:250) |
| Antibody | anti-HA (Rabbit monoclonal) | Cell Signaling Technology | Cat. No. 3724S, RRID:AB_1549585 | IF (1:500) WB (1:1000) |
| Antibody | anti-GAPDH (Mouse monoclonal) | Santa Cruz Biotechnology | Cat. No. sc-365062, RRID:AB_10847862 | WB (1:1000) |
| Antibody | anti-FHOD3 (Rabbit polyclonal) | Abcam | Cat. No. ab224463 | WB (1:1000) |
| Antibody | goat anti-rabbit IgG 800CW | LiCor Biosciences | Cat. No. 926–32211, RRID:AB_621843 | WB (1:10,000) |
| Antibody | goat anti-mouse IgG 680RD | LiCor Biosciences | Cat. No. 926–68070, RRID:AB_10956588 | WB (1:10,000) |
| Antibody | Alexa Fluor 488 goat anti-mouse | Thermo Fisher | Cat. No. A-11001, RRID:AB_2534069 | IF (1:500) |
| Antibody | Alexa Fluor 647 goat anti-rabbit | Thermo Fisher | Cat. No. A-21244, RRID:AB_2535812 | IF (1:500) |
| Recombinant DNA reagent | pGEX-6P-2-FHOD3LCT (plasmid) | This paper | | Original template, EGFP-FHOD3L, gifted by Thomas Iskratsch (*Iskratsch et al., 2010*) |
| Chemical compound, drug | Vectashield Plus Antifade mounting media | Vector Laboratories | Cat. No. H-1900–10 | |
| Chemical compound, drug | Prolong Glass Antifade mountant | Thermo Fisher | Cat. No. P36980 | |
| Chemical compound, drug | Alexa Fluor 488 Phalloidin | Thermo Fisher | Cat. No. A12379 | |
| Chemical compound, drug | Wheat Germ Agglutinin (WGA) TMR | Thermo Fisher | Cat. No. W849 | |
| Chemical compound, drug | Paraformaldehyde | Fisher Scientific | Cat. No. AC416780010 | |
| Software, algorithm | Fiji | NIH; *Schindelin et al., 2012* | | |
| Software, algorithm | MotionGUI | MATLAB; denoviral infection, driving expression of rescue constructs, *Huebsch et al., 2015* | | |
| Software, algorithm | CellPose | *Stringer et al., 2021* | | |
| Other | DAPI stain | Fisher Scientific | Cat. No. EN62248 | (1 µg/ml) |

## Protein expression, purification, and labeling

FHOD3L-CT (residues 963–1622) was cloned into pGEX-6P-2 with an N-terminal glutathione-*S*-transferase (GST) tag. The original template, EGFP-Fhod3L, was generously provided by T Iskratsch (Queen Mary University of London) (*Iskratsch et al., 2010*). Point mutations were generated by site-directed mutagenesis. Truncations were constructed using FastCloning (*Liu and Naismith, 2008*). pGEX-FHOD3L-CT GS-FH1, in which the polyproline tracts were replaced with GS linkers, was cloned via Gibson Assembly introducing a gBlock into pGEX-FHOD3L-CT (replacing the FH1 region).

The FHOD3L-CT wild-type and mutant constructs were transformed in Rosetta 2 (*E. coli* DE3) cells (Novagen), which were grown in 1 l of Terrific Broth supplemented with 100 mg/l ampicillin and 32 mg/l chloramphenicol. Expression was induced at an OD of 0.6–0.8 by adding 0.5 mM isopropyl β-D-1-thiogalacto-pyranoside (IPTG) and shaking overnight at 18°C, 210 rpm. The cells were harvested by centrifugation, washed in PBS, and flash-frozen in liquid nitrogen.

Cell pellets expressing GST-FHOD3L-CT wild-type and mutants were resuspended in 20 mM HEPES pH 7.5, 150 mM NaCl, 1 mM PMSF, 1 mM DTT, 2 µg/ml DNaseI. All subsequent steps were performed on ice or at 4°C. The cells were lysed with a microfluidizer, cleared by centrifugation at 20,000 × *g* for 20 min, and then purified using a 5 ml HitrapSP-FF cation exchange column (GE Life Sciences) with a gradient of 0.2–0.6 M NaCl over 1 column volume after a 1 column volume wash with 0.2 M NaCl. Peak fractions were dialyzed overnight into 20 mM HEPES pH 8, 200 mM NaCl, 1 mM DTT, and Prescission Protease was added to cleave off the GST-tag. This sample was centrifuged at 4°C for 48,000 rpm for 20 min and further purified on a MonoS cation exchange column (GE Life Sciences) with a gradient of 0.2–0.95 M NaCl over 40 column volumes after a 1 column volume wash with 0.2 M NaCl. Peak fractions were exchanged into storage buffer (10 mM Tris pH 8, 150 mM NaCl, 20% glycerol, 1 mM DTT), then flash-frozen in liquid nitrogen and stored at –80°C.

Cell pellets expressing GST-FHOD3S-CT were purified and stored similarly to GST-FHOD3L-CT wild-type, except 150 mM NaCl was used throughout the purification with a gradient of 0.15–0.95 M NaCl over 40 column volumes after a 1 column volume wash with 0.15 M NaCl on the MonoS cation exchange column (GE Life Sciences).

Concentrations of C-terminal FHOD3 constructs were determined by running a series of serial dilutions on a Sypro Red-stained quantitative gel using densitometry (ImageJ) with rabbit skeletal actin as the standard. All FHOD3L-CT concentrations are reported as dimer concentrations.

Human profilin-1 and *S. pombe* profilin were expressed and purified as described for *Drosophila* profilin (Chic) (**Bor et al., 2012**). Profilin-1 concentration was determined using the extinction coefficient of 14,992 $M^{-1}$ $cm^{-1}$. *S. pombe* profilin concentration was determined using 1.63 OD/mg/ml (**Lu and Pollard, 2001**).

We used RSA throughout the paper, based on FHOD3L's role in skeletal and cardiac muscle. Skeletal muscle actin was isolated from rabbit back muscle acetone powder (Pel-Freez) according to the method described by Spudich and Watt, followed by gel purification (**Spudich and Watt, 1971**). Skeletal muscle actin was labeled with pyrene iodoacetamide (Thermo Scientific) or Alexa Fluor 488 NHS-ester (Thermo Scientific) as described (**Sun et al., 2018**).

## Pyrene assays

Pyrene assays were performed essentially as described (**Bor et al., 2012**) on an Infinite 200 Pro plate reader (Tecan). FHOD3L-CT was diluted in buffer Z (2 mM Tris pH 8.0, 0.2 mM ATP, 0.1 mM $CaCl_2$, 0.5 mM TCEP, 0.04% sodium azide) before addition to polymerization buffer (KMEH: 10 mM HEPES, pH 7, 1 mM EGTA, 50 mM KCl, 1 mM $MgCl_2$). This mix was added to $Mg^{2+}$-actin at a final concentration of 4 µM with 5% pyrene-labeled actin. For bulk assembly assays, nucleation strengths were calculated from the slope at $t_{1/8}$. For seeded elongation assays, actin filaments were sheared by passing three times through a 24-gauge needle and then aliquoted into each well of a microplate. Proteins were added to the seeds and incubated for 2–4 min at room temperature. Seeds and additional proteins in KMEH were added to $Mg^{2+}$-actin, at a final concentration of 0.5 µM actin with 10% pyrene-labeled actin, to initiate elongation. Elongation rates were determined by linear regression over the first 90 s and normalized against the rate of actin alone in each experiment. The affinity of FHOD3-CT for barbed ends was determined by fitting the data to the quadratic binding equation

$$r = a + b * \left( ([\text{barbed ends}] + [\text{FHOD3-CT}] + K_d) - \sqrt{([\text{barbed ends}] + [\text{FHOD3-CT}] + K_d)^2 - 4 * [\text{barbed ends}] * [\text{FHOD3-CT}]} \right),$$

where *r* is the normalized elongation rate, and *a* and *b* are offset and scaling constants, respectively.

## Low-speed bundling assays

10 µM actin was polymerized for 1 hr at room temperature and diluted to 5 µM with varying amounts of FHOD3-CT constructs in KMEH. Samples were incubated for 30 min at room temperature and then spun for 20 min, 14,000 × *g* to separate the pellet and supernatant. The supernatant samples were carefully transferred to new tubes, and the pellet samples were resuspended in an equal volume of 1× sample loading buffer for quantitative comparison. Samples were boiled in 1× sample loading buffer

for 10 min and run on 10% SDS-PAGE gels. The percentage of actin pelleted was determined via densitometry with Fiji (*Schindelin et al., 2012*).

## TIRF microscopy

TIRF microscopy was utilized to measure the elongation rates and run lengths of the FHOD3-CT constructs. Coverslips were rinsed three times in MilliQ water, placed in 2% Hellmanex (Hellma Analytics) at 60–65°C for 2 hr, rinsed another five times in MilliQ water, and allowed to dry.

Parallel flow chambers of ~15 µl were assembled on the slide using strips of double-sided tape. Flow chambers were prepared with the following steps: (1) 20 µl of high-salt buffer (50 mM Tris pH 7.5, 600 mM NaCl) for 2 min; (2) 12.5 µl of 60 nM NEM-myosin (stock in 10 mM HEPES pH 7, 0.5 M KCl, 10 mM EDTA pH 7.0, 1 mM DTT, 50% glycerol, diluted in high-salt buffer); (3) 20 µl of high-salt BSA-containing buffer (1% BSA, 50 mM Tris pH 7.5, 600 mM NaCl); (4) 20 µl of low-salt BSA-containing buffer (1% BSA, 50 mM Tris pH 7.5, 150 mM NaCl); (5) 50 µl of magnesium-actin and additional proteins to be assayed for a final concentration of 1 µM actin (10% Alexa Fluor 488-labeled) and 5 µM human profilin-1 in 1× TIRF buffer (KMEH, 0.2 mM ATP, 50 mM DTT, 20 mM glucose, 0.5% methylcellulose [4000 cP]) supplemented with 250 µg/ml glucose oxidase and 50 µg/ml catalase.

The videos and images were acquired using a Zeiss Axio Observer 7 Basic Marianas Microscope with Definite Focus 2 equipped with a 3i Vector TIRF System, an Alpha Plan-Apochromat ×63 (1.46 NA) Oil TIRF Objective, and an Andor iXon3 897 512×512 10 MHz EMCCD Camera, using Slidebook 6 software. Experiments were performed at room temperature. Images were captured at 2.5 s intervals for 10 min. Filament lengths were quantified with the JFilament plug-in in Fiji (*Schindelin et al., 2012*; *Smith et al., 2010*). Bright and dim filaments were distinguished manually. Due to the transient nature of the dim portions of filaments generated by FHOD3L constructs, pauses for actin elongation were first visually identified as portions of roughly 0 slope deviating away from the best fit line off the subunits added vs. time plots of elongation. The corresponding times for the duration of the pauses were then more carefully examined. Events were defined as lasting at least 5 s, with slopes within –10 and 5. These pauses almost always preceded a burst of elongation by either FHOD3S/L-CT WT or K1193. To estimate the elongation rate, the dim region growth was fit to a line. It was confirmed that the elongation rate reduced back to that of profilin-actin alone when the filament intensity increased. The $R^2$ value of these bursts of elongation was all >0.5, but in most cases >0.8.

## Adenoviral generation, purification, and infection

We used HEK 293 cells solely for the production of adenoviral vectors. The HEK 293 cell line was obtained from the lab of Dr. Kohnosuke Mitani at UCLA. Dr. Mitani's lab was one of the pioneering groups in the development of adenoviral vectors for gene therapy and maintained high standards of quality control for cell culture and maintenance. Because the success of adenoviral vector production heavily depends on the quality of HEK 293 cells, we relied on the robustness of the original cell source. Although we have not performed formal authentication of the cell line since the acquisition, we have strictly followed the protocols established by Dr. Mitani's lab. This includes using cells with low passage numbers and closely monitoring cell morphology and functionality. The cells are cryopreserved in liquid nitrogen and maintained under controlled conditions.

Adenoviruses containing a full-length FHOD3L wild-type construct were generated by transiently transfecting HEK293 cells using the CalPhos Mammalian Transfection Kit (Catalog No. 631312, Takara Bio) with pAdenoX-CMV-3xHA-FHOD3L-CMV-DsRed after digesting with PacI (NEB) in Cutsmart buffer (NEB) for 1 hr at 37°C. HEK293 cells were cultured in 1X DMEM (Thermo Fisher, Catalog No. 11965092), 10% FBS (Thermo Fisher, Catalog No. 10082147), and 1% penicillin/streptomycin/amphotericin B (Thermo Fisher, Catalog No. 15240096). Cells were lysed 7–10 days later when noticeable cytopathic effect was present and centrifuged at 1.5k × *g* for 5 min at room temperature.

Additional crude adenovirus encoding full-length FHOD3L was harvested after infecting multiple 6 cm dishes of HEK293. Dilutions and subsequent infections were performed in PBS with magnesium chloride and calcium chloride (PBS+/+) (Thermo Fisher, Catalog No. 14040117) for 1 hr at room temperature, rotating the plate every 10–15 min. Plaque formation assays were then performed as in *Baer and Kehn-Hall, 2014*. Individual viral plaques were collected with a P1000 tip to isolate adenoviral clones of full-length FHOD3L. Clone variability was assessed by infecting HEK293 for each adenoviral clone and observing subsequent DsRed expression over time. Optimal adenoviral clones

were selected for large-scale purification. Five 15 cm dishes of HEK293 were infected, then collected 2–3 days later for purification with the Adeno-X Maxi Purification Kit (Takara Bio, Catalog No. 631533).

Crude adenoviruses containing pAV-CMV-{3xHA-FHOD3L GS-FH1}:SV40 pA-CMV-mCherry and pAV-CMV-{3xHA-FHOD3L K1193L}:SV40 pA-CMV-mCherry were commercially generated and purchased from VectorBuilder. The amount of crude adenovirus was estimated by infecting serially diluted amounts of crude adenovirus in a six-well plate of HEK293 and then used to infect five 15 cm dishes of HEK293 for 1 hr at room temperature and collected 2–3 days later for purification with the Adeno-X Maxi Purification Kit (Takara Bio, Catalog No. 631533).

Particle titers of purified viruses were quantified by making three separate dilutions of each virus in 0.1% SDS (Thermo Fisher) and vortexing for 5 min, spinning down 13,000× rpm, 5 min, and then taking the average of the absorbance readings on the Nanophotometer N50-GO (Implen) to quantify particles/ml as per *Maizel et al., 1968*.

## NRVM seeding, siRNA knockdown, and rescue

NRVMs were isolated from postnatal P1- to P3-day-old Sprague-Dawley rat pups of mixed gender by UCLA Cardiovascular Research Theme Core services. The ethical approval for use was obtained from the Animal Research Committee (ARC) for protocol 2008-126. Eight-well chamber slides (Corning, Catalog No. 354118) were prepared by coating with 10 mg/ml fibronectin (Sigma, Catalog No. F1141) and 20 mg/ml poly-D-lysine hydrobromide (Sigma, Catalog No. P6407) in PBS overnight at 4°C. NRVMs were seeded at 175,000 NRVMs/well in a mixture of 75% DMEM (Thermo Fisher, Catalog No. 11965092), 15% Medium-199 (Thermo Fisher, Catalog No. 11150059) supplemented with 2 mM L-glutamine (Thermo Fisher, Catalog No. A2916801) and 10 mM HEPES (Thermo Fisher, Catalog No. 15630080). For reverse transfection, the cells were treated with siRNA targeting FHOD3 (QIAGEN, Rn_LOC100360334_2 Flexitube siRNA) or AllStars Negative Control siRNA (QIAGEN, 20 nmol), Lipofectamine RNAiMAX transfection reagent (Thermo Fisher, Catalog No. 13778075) and Opti-MEM I Reduced Serum Medium (Thermo Fisher, Catalog No. 31985062) to dilute the siRNA. Media was changed 24 hr later to one containing penicillin/streptomycin (Thermo Fisher, Catalog No. 15140122). NRVMs were infected with an optimal amount of adenovirus (determined experimentally to be particle titer MOI 350 to minimize excessive damage to NRVMs while maintaining sufficient expression) as above 48 hr after seeding, and media containing penicillin/streptomycin was added on top of the PBS+/+ at the end of the infection. Media was changed 24 hr later, and cells were examined 48 hr after infection.

## Gene expression analysis by quantitative reverse-transcriptase PCR

RNA was extracted from the NRVMs 4 days after reverse transfection of siRNA using the Direct-zol RNA mini prep kit (Zymo Research, Catalog No. R2050). RNA was reverse-transcribed into complementary DNA using the qScript cDNA synthesis kit (Quanta Biosciences, Catalog No. 95047-025). Quantitative reverse-transcriptase PCR was performed using PowerUp SYBR green master mix for qPCR (Applied Biosystems, Catalog No. A25742) on a Lightcycler 480 (Roche). Each qPCR was repeated three times. Forward and reverse primer sequences are as follows: GAPDH forward, CCGCATCTTCTTGTGCAGTG; GAPDH reverse, CGATACGGCCAAATCCGTTC; FHOD3 forward, CAGCCAATCACGGAG; FHOD3 reverse, TGCTGTCCTTGCCCTGA.

## Western blots

NRVMs were lysed in 100 mM Tris pH 8, 150 mM NaCl, 0.5% Triton-X from eight-well chamber slides (Corning, Catalog No. 354118) after rescue experiments, and samples were vortexed for 1 min before centrifuging at 15,000 rpm, 4°C, for 10 min. The resulting supernatant was boiled in sample loading buffer at 100°C for 10 min and run on an SDS-PAGE gel. The gel was transferred to an Immobilon-FL polyvinylidene fluoride membrane (Millipore, IPFL00010) at 100 V for 90 min on ice. The membrane was blocked in 4% nonfat milk in low-salt TBST (20 mM Tris pH 7.6, 150 mM NaCl, 0.05% Tween-20, 0.01% sodium azide) for 30 min at room temperature. It was incubated, rotating at 4°C overnight, with Fhod3 polyclonal rabbit (Abcam, ab224463) or HA monoclonal rabbit (Cell Signaling Technologies, 3724S) and GAPDH monoclonal mouse (Santa Cruz Biotechnology, sc-365062) diluted 1:1000 in low-salt TBST. Membranes were washed three times for 5 min each the next day in high-salt TBST (20 mM Tris pH 7.6, 500 mM NaCl, 0.05% Tween-20, 0.01% sodium azide) and then incubated for 1 hr at room

temperature, shaking, with 800CW goat anti-rabbit IgG secondary antibody (Li-Cor Biosciences, 926-32211) and 680RD goat anti-mouse IgG secondary antibody (Li-Cor Biosciences, 926-68070) diluted 1:10,000 in high-salt TBST. Membranes were washed three times for 10 min, each in high-salt TBST and then imaged on a Li-Cor Odyssey 9120 Infrared Imager (Li-Cor Biosciences).

## In vitro contractility assay

Contractility assessments were performed by utilizing a video-based technique with the UCSF Gladstone-developed MATLAB program MotionGUI (*Huebsch et al., 2015*). Videos for contractility analysis were acquired using MicroManager software on a Leica SD AF Spinning Disc system using an HC PL Fluotar 10× (0.3 NA) dry objective lens with an ORCA-Flash4.0 LT C11440 camera (Hamamatsu) at 30 fps with live NRVMs incubated at 37°C, 5% $CO_2$ (Tokai Hit). The videos were converted from ome.tif to a tiff stack with Fiji for analysis with MotionGUI. A pixel size of 0.538 µm was obtained from the metadata. Eight-pixel macroblocks were used for all assessments. All parameters of the MotionGUI program not specified here were set to their respective default values. Motion vectors were calculated, and the data were evaluated upon completion. All videos were subjected to the same post-processing procedures to ensure consistency during comparative analysis. Each video sample was post-processed using neighbor-based cleaning with the vector-based cleaning criterion within the program. The threshold for this post-processing method was set to two for all samples and was adequate for improving the signal-to-noise ratio enough to identify peaks clearly corresponding to beating events in most samples.

## IF and image analysis

For sarcomere integrity analysis, NRVMs were stained before fixation with WGA TMR (Thermo Fisher, Catalog No. W849) for 10 min at 37°C with 5 µg/ml WGA, followed by two PBS washes. NRVMs were then fixed with 4% paraformaldehyde (Fisher Scientific, Catalog No. AC416780010) in PBS for 15 min at 37°C. Cells were washed three times in PBS for 5 min, each at room temperature. Cells were permeabilized and blocked with PBS/0.1% Triton-X/10% Goat Serum (Triton-X: Thermo Fisher, Catalog No. A16046.AP; Goat Serum: Sigma, Catalog No. S26-100ML) for 30 min at 37°C. Cells were incubated with primary antibodies overnight at 4°C. α-Actinin (mouse) primary antibody (Sigma, Catalog No. A7811) was diluted 1:250, and HA-tag (rabbit) primary antibody (Cell Signaling, Catalog No. 3724) was diluted 1:500 in PBS/0.1% Triton-X/5% Goat Serum. Cells were washed three times for 5 min, each with PBS/0.1% Triton-X/5% Goat Serum at room temperature. Incubation with secondary antibodies was for 1 hr at 37°C. Alexa Fluor 488 goat anti-mouse secondary (Thermo Fisher, Catalog No. A-11001) or Alexa Fluor 647 goat anti-rabbit secondary (Thermo Fisher, Catalog No. A-21244) was diluted 1:500 in PBS/0.1% Triton-X/5% Goat Serum. Cells were then washed twice in PBS/0.1% Triton-X/5% Goat Serum for 5 min each, followed by one PBS wash for 5 min before mounting in Vectashield Plus Antifade mounting media (Vector Laboratories, Catalog No. H-1900-10) with 1 µg/ml DAPI (Fisher Scientific, Catalog No. EN62248).

For thin filament determination, NRVMs were incubated with sterile-filtered Ringer's relaxation buffer (6 mM potassium phosphate pH 7.0, 100 mM NaCl, 2 mM KCl, 0.1% glucose, 2 mM $MgCl_2$, 1 mM EGTA) for 20 min, room temperature, followed by a brief PBS wash to start the WGA staining. Fixation and blocking were performed as above. Staining with HA-tag (rabbit) primary antibody and Alexa Fluor 647 goat anti-rabbit secondary antibody was performed as above. Cells were then washed twice in PBS/0.1% Triton-X/5% Goat Serum for 5 min each, and 200 nM Alexa Fluor 488 Phalloidin (Thermo Fisher, Catalog No. A12379) was added for 1 hr, room temperature in PBS/1% BSA. Cells were washed with PBS twice for 5 min before mounting in Prolong Glass Antifade Mountant (Thermo Fisher, Catalog No. P36980) with 1 µg/ml DAPI (Fisher Scientific, Catalog No. EN62248).

Twelve ×40 magnification images per biological replicate were acquired using a random coordinate generator (https://onlineintegertools.com/generate-integer-pairs), excluding the edges of the wells, for quantification of sarcomeres and thin filaments. Sample size with sufficient power was determined via a small-scale pilot study for sarcomere number, length, and width, as well as thin filaments between mock knockdown, mock rescue, and FHOD3L WT rescue, followed by inputting averages of the treatments on https://www.stat.ubc.ca/~rollin/stats/ssize/n1.html. Each well was treated as a 16×16 grid of ×40 magnification fields of view. 16-bit images were acquired on an AXIO Imager.D1 fluorescence phase contrast microscope (Zeiss) using an EC Plan-Neofluar 40x M27 (0.75 NA) objective lens with an

AxioCam MRm camera (Zeiss). Images of NRVMs in figures were acquired with a Plan-Apochromat oil DIC M27 63× (1.4 NA) objective lens for visualization.

Images showing DsRed reporter fluorescence at the end of the rescue timeline for the NRVMs were taken on a CKX53 inverted phase contrast microscope (Olympus) using a UPLFLN 4× (0.2 NA) objective lens (Olympus) with an ORCA-spark digital CMOS camera (Hamamatsu).

Cells were segmented using CellPose (*Stringer et al., 2021*). Two-color images of the WGA membrane stain and nuclear stain were saved as jpeg files on Fiji. These images were then input to the CellPose environment for batch processing with a cell diameter of 210 pixels, 0.4 cell probability and flow thresholds, 0 stitch threshold, using the cyto2 model for segmentation. The text files of the outlines were saved and overlaid on the respective channels of the images individually using the provided Python file from CellPose as a macro through Fiji (imagej_roi_converter.py). CellPose segments with no nuclei or three or more nuclei, as well as cell segments on the edge of the image, were excluded from analysis.

Sarcomere analysis was performed manually in a single-blind manner (for mock knockdown, mock rescue, wild-type rescue, and the K1193L rescue) using blindrename.pl (https://github.com/davalencia0914/sarcApp_Cellpose_Merge, copy archived at *Valencia, 2025*) to generate the filenames and a key filename csv file for decoding after analysis. We attempted to blind all other rescue conditions, but they were too easily identified based on expression level and localization differences. Linescans were generated along myofibrils, perpendicular to the Z-lines, to make sarcomere length measurements from Z-line peak to Z-line peak. Z-line lengths were measured by visual inspection of the α-actinin channel and measured on Fiji with the line tool. Three or more consecutive Z-lines, at least 0.70 µm long in a row, were analyzed as sarcomeres.

## Statistical analysis

To compare two or more groups for the rescue experiments in NRVMs, pair-wise comparisons were performed. In order to reduce the Type 1 error rate stemming from multiple comparisons, Bonferroni correction was applied to obtain a corrected alpha. Whether groups were normally distributed or not was determined by the Shapiro-Wilk test. To compare two normally distributed groups, Student's two-sample, unpaired t-test was used. If either group was not normally distributed, the nonparametric Mann-Whitney U test was used. The same analysis was applied to analyze run lengths and capping duration from the TIRF seeded elongation assays.

For the nucleation assay, barbed-end binding assays, and TIRF elongation rate analysis, one-way ANOVAs were performed for three treatment comparisons with post hoc Tukey tests as the variances between all treatments tested were fairly equal.

In all cases when the sample size exceeded 20, we removed outliers from the bottom and upper 2.5% of data to perform all statistical tests, regardless of normality, to avoid introducing bias. However, all plots shown include these outliers.

## Acknowledgements

We thank members of the Quinlan lab for experimental support and scientific feedback, the Reisler lab for valuable discussions, experimental support, and reagents, the Courtemanche lab for NEM-myosin, and T Iskratsch for the FHOD plasmids. We also thank Dr. Michael D Roth for access to his lab's BSL II-equipped facilities, enabling the production of adenovirus. This work was supported by NIH grant R01 HL146159 to MEQ, and Ruth L Kirschstein National Research Service Award T32 GM007185 to DAV and T32 AR065972 to ANK. Confocal laser scanning microscopy was performed at the Advanced Light Microscopy/Spectroscopy Laboratory and Leica Microsystems Center of Excellence at the California NanoSystems Institute at UCLA (RRID:SCR_022789) with funding support from NIH Shared Instrumentation Grant S10OD025017 and NSF Major Research Instrumentation grant CHE-072251.

## Additional information

### Funding

| Funder | Grant reference number | Author |
| --- | --- | --- |
| National Institutes of Health | R01 HL146159 | Margot E Quinlan |
| National Institutes of Health | T32 GM007185 | Dylan A Valencia |
| National Institutes of Health | T32 AR065972 | Angela N Koeberlein |

The funders had no role in study design, data collection and interpretation, or the decision to submit the work for publication.

### Author contributions

Dylan A Valencia, Formal analysis, Investigation, Methodology, Writing – original draft, Writing – review and editing; Angela N Koeberlein, Formal analysis, Investigation, Writing – review and editing; Haruko Nakano, Formal analysis, Supervision, Methodology, Project administration, Writing – review and editing; Akos Rudas, Formal analysis, Writing – review and editing; Aanand A Patel, Conceptualization, Investigation; Airi Harui, Resources, Supervision, Methodology, Writing – review and editing; Cassandra Spencer, Formal analysis; Atsushi Nakano, Conceptualization, Funding acquisition, Methodology, Project administration, Writing – review and editing, Supervision; Margot E Quinlan, Conceptualization, Data curation, Formal analysis, Funding acquisition, Methodology, Project administration, Writing – review and editing

### Author ORCIDs

Dylan A Valencia ⬤ https://orcid.org/0000-0003-1503-5081
Haruko Nakano ⬤ https://orcid.org/0000-0001-5807-9127
Aanand A Patel ⬤ https://orcid.org/0000-0002-1509-9950
Airi Harui ⬤ https://orcid.org/0000-0002-5730-4035
Atsushi Nakano ⬤ https://orcid.org/0000-0001-5702-5039
Margot E Quinlan ⬤ https://orcid.org/0000-0002-8133-1033

### Ethics

The ethical approval of rat cardiomyocyte use was obtained from the Animal Research Committee (ARC) for protocol 2008-126.

Reviewer #1 (Public review): https://doi.org/10.7554/eLife.104048.3.sa1
Reviewer #3 (Public review): https://doi.org/10.7554/eLife.104048.3.sa2
Author response https://doi.org/10.7554/eLife.104048.3.sa3

## Additional files

### Supplementary files

MDAR checklist

### Data availability

All gels and western blots analysed during this study are included in the article and supporting files; source data files have been provided for Figures 1, 2, and 4. All code used can be found on GithHub (https://github.com/davalencia0914/sarcApp_Cellpose_Merge; copy archived at *Valencia, 2025*).

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
