## [Editor Report · eLife Assessment]

Valencia et al. combine elegant in vitro biochemical experiments with functional assays in cardiomyocytes to determine which properties of the FHOD3 formin are essential for sarcomere assembly. Using separation-of-function mutants, they show that FHOD3's elongation activity, rather than its nucleation, capping, or bundling activities, is key to its sarcomeric function. This is an **important** finding and the data presented in the manuscript are **convincing**; however, the presence of FHOD3 at filament barbed ends in the TIRF elongation assays should probably be verified directly in a future study.

---

## [Referee Report · Reviewer #1 (Public review)]

Summary:

Formins are complex proteins with multiple effects on actin filament assembly, including nucleation, capping with processive elongation, and bundling. Determining which of these activities are important for a given biological process and normal cellular function is a major challenge.

Here, the authors study the formin FHOD3L, which is essential for normal sarcomere assembly in muscle cells. They identify point mutants of FHOD3L in which formin nucleation and elongation/bundling activities are functionally separated. Expression of these mutants in neonatal rat ventricular myocytes shows that the control of actin filament elongation by formin is the major activity required for normal assembly of functional sarcomeres.

Strengths:

The strength of this work is to combine sensitive biochemical assays with excellent work in neonatal rat ventricular myocytes. This combination of approaches is highly effective for analyzing the function of proteins with multiple activities in vitro. The authors have pushed the experiments and data analysis as far as possible with the technologies available to them.

Weaknesses:

FHOD3L is not the easiest formin to study because of its relatively weak nucleation activity and the short duration of capping events. This difficulty imposes rigorous biochemical analysis and careful interpretation of the data. As the authors acknowledge, it will be important in future to perform complementary multi-color TIRF experiments to confirm that the brief accelerations in the elongation of actin filaments are indeed due to FHOD3 binding.

---

## [Referee Report · Reviewer #3 (Public review)]

Valencia et al. aim to elucidate the biochemical and cellular mechanisms through which the human formin FHOD3 drives sarcomere assembly in cardiomyocytes. To do so, they combined rigorous in vitro biochemical assays with comprehensive in vivo characterizations, evaluating two wild type FHOD3 isoforms and two function-separating mutants. Surprisingly, they found that both wild type FHOD3 isoforms can nucleate new actin filaments, as well as elongate existing actin filaments in conjunction with profilin following barbed-end capping. This is in addition to FHOD3's proposed role as an actin bundler. Next, the authors focused on the longer isoform FHOD3L due to its essential role in sarcomere assembly in cardiomyocytes. They asked whether FHOD3L promote sarcomere assembly through its activity in actin nucleation or rather elongation. To do so, the authors designed two function-separating mutants: the K1193L mutation in the FH2 domain, known for its importance in actin nucleation, and the glycine-serine linker substitution in the FH1 domain ("GS-FH1",) known for its requirement in actin elongation. They demonstrated that while K1193L maintains its elongation activity and greatly diminishes nucleation and bundling, in GS-FH1 keeps its nucleation activity while lose its capacity to drive elongation. Armed with these tools, the authors attempted to rescue FHOD3L siRNA-treated neonatal rat ventricular myocytes (NRVM) with transgenes carrying wild type, K1193L, or GS-FH1 mutant forms of human FHOD3. In each condition, they evaluated the numbers and morphology of sarcomeres, as well as their ability to beat and generate cardiac rhythm. The authors found that while the wild type FHOD3L and the K1193L mutant can rescue sarcomere morphology and physiology, the GS-FH1 mutant fails to do so. Given that in GS-FH1 mainly elongation activity is compromised, the authors concluded that the elongation activity of FHOD3 is essential for its role in sarcomere assembly in cardiomyocytes, while its nucleator activity is dispensable. Overall, this important study provided a broadened view on the biochemical activities of FHOD3, and a pioneering view on a possible cellular mechanism of how FHOD3L drives sarcomere assembly. If further validated, this can lead to new mechanistic models of sarcomere assembly and potentially new therapeutic targets of cardiomyopathy.

The conclusions of this paper are mostly well supported by the comprehensive biochemical analyses performed by the authors. In my original assessment, I raised the point that the extreme low level of GS-FH1 signal in transfected cells in Figure 6A may reflect a failure of actin-binding by this construct in vivo, rather than its inability of driving elongation. The authors have thoroughly addressed this concern by: (1) providing new images of the GS-FH1 rescue condition with HA-FHOD3L signal intensities matching that of the K1193L rescue condition, and (2) quantitatively demonstrating that the expression levels in the GS-FH1 rescue condition are comparable with that of wild type FHOD3L rescue condition. This is nicely complemented by the new phalloidin staining of the GS-FH1 rescue condition, which showcased additional details of actin puncta reminiscent of that present in muscle stress fibers or premyofibrils. Overall, I am now convinced that the GS-FH1 cannot rescue sarcomere formation even when expressed at comparable levels. Given that GS-FH1 demonstrates actin elongation defects in vitro, it is reasonable to conclude that the actin elongation function of FHOD3L is essential for sarcomere formation in vivo.

---

## [Author Response]

The following is the authors’ response to the original reviews.

**Public Reviews:**

**Reviewer #1 (Public review):**
Summary:Formins are complex proteins with multiple effects on actin filament assembly, including nucleation, capping with processive elongation, and bundling. Determining which of these activities is important for a given biological process and normal cellular function is a major challenge.Here, the authors study the formin FHOD3L, which is essential for normal sarcomere assembly in muscle cells. They identify point mutants of FHOD3L in which formin nucleation and elongation/bundling activities are functionally separated. Expression of these mutants in neonatal rat ventricular myocytes shows that the control of actin filament elongation by formin is the major activity required for the normal assembly of functional sarcomeres.Strengths:The strength of this work is to combine sensitive biochemical assays with excellent work in neonatal rat ventricular myocytes. This combination of approaches is highly effective for analyzing the function of proteins with multiple activities in vitro.Weaknesses:FHOD3L does not seem to be the easiest formin to study because of its relatively weak nucleation activity and the short duration of capping events. This difficulty imposes rigorous biochemical analysis and careful interpretation of the data, which should be improved in this work.

We thank the reviewer for their praise and appreciation of our work. Indeed, FHOD3L is a challenging formin to work with.

Important points are raised here and below regarding the brief elongation events we reported. As suggested, we performed more rigorous analysis of the data and present it in the revised manuscript. We now report that from 45 dim regions analyzed, in three independent experiments with wild type FHOD3L, we detected 40 bursts. (The remaining five could be formin falling off too quickly to detect or the dim spots could be regions of inhomogeneity in intensity, not due to formin.) For comparison to the presented data with FHOD3L-CT, we analyzed the filaments in TIRF assays with no formin present. As the reviewers point out, inhomogeneities in filament intensity are normal. Thus, we examined any dim spots for pauses and/or bursts. As is now reported in Figure 2G,H, the velocity of growth of these dim spots is indistinguishable from the velocity of the rest of the filament. We acknowledge that our numbers may not be perfectly accurate, due to the noise in our system, we believe that the difference of 3-4 fold increase versus no change in rate is substantial and convincing.

We also determined the number of dim spots per length of filament. We found a higher frequency when FHOD3L-CT or FHOD3S-CT was present vs no formin, as now shown in Figure 2 – supplements 1G and 2E.

We were asked about the pauses we observe before bursts of elongation and how we know they are functionally relevant. The short answer is that we do not know. We reported them because they were so common: Of the 40 bursts, pauses preceded the burst in 38 cases. We cannot rule out that this pause reflects an interaction with the surface but might expect the frequency to be lower if it were. We revise the text to make our conclusions about pauses more circumspect.

We are convinced that the brief dim events we observed in the presence of FHOD3L-CT, in fact, reflect formin-mediated elongation and worked hard to improve their presentation, in addition to the added analysis. We include new kymographs, including examples from FHOD3L, FHOD3S, K1193L, and actin alone. We hope that the reviewers are also convinced.

This does not preclude our interest in the microfluidics and two-color assays, which will be pursued in the future. We have reached out to a colleague who is set up to repeat these measurements with microfluidics-assisted TIRF. The noise should be greatly reduced and the system is also optimal for directly visualizing labeled FHOD3, as suggested. We expect these experimental approaches will provide additional insights.

**Reviewer #2 (Public review):**
This article elucidates the biochemical and cellular mechanisms by which the FHOD-family of formins, particularly FHOD3, contributes to sarcomere formation and contractility in cardiomyocytes. Formins are mainly known to nucleate and elongate actin filaments, with certain family members also exhibiting capping, severing, and bundling activities. Although FHOD3 has been well-established as essential for sarcomere assembly in cardiomyocytes, its precise biochemical functions and contributions to actin dynamics remain poorly understood.In this study, the authors combine in vitro biochemical assays with cellular experiments to dissect FHOD3's roles in actin assembly and sarcomere formation. They demonstrate that FHOD3 nucleates actin filaments and acts as a transient elongator, pausing elongation after an initial burst of filament growth. Using separation-of-function mutants, they show thatFHOD3's elongation activity - rather than its nucleation, capping, or bundling capabilities - is key for its sarcomeric function.The experiments have been conducted rigorously and well-analyzed, and the paper is clearly written. The data presented support the authors' conclusions. I appreciate the detailed description and rationale behind the FHOD3 constructs used in this study.

We are happy to hear others find paper to be clearly written and well described.

However, I was somewhat surprised and a bit disappointed that while the authors conducted single-color TIRF experiments to observe the effects of FHOD3 on single filaments, they did not use fluorescently labeled FHOD3 to directly visualize its behavior. Incorporating such experiments would significantly strengthen their conclusions regarding FHOD3's bursts of elongation interspersed with capping activity. While I understand this might require a few additional weeks of experiments, these data would add considerable value by directly testing the proposed mechanism.

We appreciate the suggestion and hope to incorporate a two-color approach soon. As noted, FHOD3L is not always easy to work with and we do not have a functional labeled copy of the protein at this time.

There is a typo in the word "required" in line number 30. The authors also use fit data to extract parameters in several panels (e.g., Figures 2b, 2d, 3a, and 3b). While these fit functions may be intuitive to actin experts, explicitly describing the fit functions in the figure legends or methods would greatly benefit the broader readership.

Thank you for these comments. We updated the indicated figures and described the analysis in greater detail.

**Reviewer #3 (Public review):**
Valencia et al. aim to elucidate the biochemical and cellular mechanisms through which the human formin FHOD3 drives sarcomere assembly in cardiomyocytes. To do so, they combined rigorous in vitro biochemical assays with comprehensive in vivo characterizations, evaluating two wild-type FHOD3 isoforms and two function-separating mutants. Surprisingly, they found that both wild-type FHOD3 isoforms can nucleate new actin filaments, as well as elongate existing actin filaments in conjunction with profilin following barbed-end capping. This is in addition to FHOD3's proposed role as an actin bundler. Next, the authors asked whether FHOD3L promotes sarcomere assembly in cardiomyocytes through its activity in actin nucleation or rather elongation. With two function-separating mutants, the authors evaluated the numbers and morphology of sarcomeres, as well as their ability to beat and generate cardiac rhythm. The authors found that while the wild-type FHOD3L and the K1193L mutant can rescue sarcomere morphology and physiology, the GS-FH1 mutant fails to do so. Given that in GS-FH1 mainly elongation activity is compromised, the authors concluded that the elongation activity of FHOD3 is essential for its role in sarcomere assembly in cardiomyocytes, while its nucleator activity is dispensable. Overall, this important study provided a broadened view on the biochemical activities of FHOD3, and a pioneering view on a possible cellular mechanism of how FHOD3L drives sarcomere assembly. If further validated, this can lead to new mechanistic models of sarcomere assembly and potentially new therapeutic targets of cardiomyopathy.The conclusions of this paper are mostly well supported by the comprehensive biochemical analyses performed by the authors. However, the sarcomere assembly defect phenotype in the GS-FH1 rescue condition requires further investigation, as the extremely low level of GS-FH1 signal in transfected cells in Figure 6A may reflect a failure of actin-binding by this construct in vivo, rather than its inability to drive elongation. Though the authors do show in Figure 6 that GS-FH1 can bind to normal-looking sarcomeres when they are present, this may be due to a lack of siRNA activity in these cells, such that endogenous FHOD3L is still present. In this possible scenario, GS-FH1 may dimerize with endogenous FHOD3L. The authors should demonstrate that GS-FH1 alone can indeed interact with existing actin filaments in vivo. While this has been clearly demonstrated in vitro, given the more complex biochemical environment in vivo where additional unknown binding partners may present, cautions should be made when extrapolating findings from the former to the latter.

The reviewer is concerned about the low protein levels in the GS-FH1 rescue experiments as reflected in the HA fluorescence intensity distributions shown in Fig. 5 Supplement 2A. While the scenario proposed could explain our observations with the GSFH1 rescues it is quite complex. Nor does the scenario preclude the conclusion that the FH1 domain is critical. We agree that the observed sarcomeres are likely to be residual in cells with incomplete RNAi. We now include the image of a cell that is still full of sarcomeres and note that the GH-FH1 is expressed at a relatively high level and striated throughout the cell. We interpret this as evidence that GS-FH1 is stable when suitable binding sites are available. We cannot exclude that there is more GS-FH1 because there was more endogenous FHOD3L with which to heterodimerize. If the GS-FH1 heterodimer were simply poisoning the wild type protein, we do not expect that it would be bound correctly to sarcomeres. If, instead, heterodimers have some activity, it seems far from sufficient to rescue sarcomere formation, suggesting that two functional FH1 domains are critical.

Furthermore, we do not see evidence of correlation between protein levels and rescue at the level present in these cells (addressed below). Unfortunately, the proposed IP to test whether FHOD3L binds actin in vivo would only potentially report on filament side binding (both direct and indirect). It would not address whether the GS-FH1 mutant functions as a nucleator, elongator, bundler and/or capping protein in vivo.

The critical question that we can address is whether the phenotype is due to low protein levels, assuming the protein present is functional, or due to loss of elongation activity by FHOD3L. To address this question, we returned to our data.

First, we plotted the distributions of the intensities of the cells we analyzed further, in addition to the automated readout of all of the cells in the dish (Fig. 4 supplement 1). These cells were selected randomly and, as should be the case, the distributions of their intensities agree well with the original distributions for the three different rescue constructs: FHOD3L, K1193L, and GS-FH1 (Fig. 6 supplement 1). We then asked whether there was any correlation in HA intensities with the sarcomere metrics. As seen in our pilot data, no correlation is evident in any of the three cases across the range of intensities we collected (400 – 2700 a.u.) (old Fig. 6 supplement C,D,E). We now replace the data from pilot experiments with analysis of HA intensities and sarcomere metrics from the data sets included in the paper (new Fig 6. Supplement 1). Again, little to no correlation was observed (the single highest r-squared value is 0.2 and the remaining eight values are less than or equal to 0.08).

To more specifically address the question of whether low HA fluorescence intensity is likely to reflect sufficient protein levels to build sarcomeres we re-examined two data sets from the FHOD3L WT rescue data. We found that, by chance, the first replicate of data from the wild type rescue has a comparable intensity distribution to that of the GSFH1 rescues (580 +/- 261 / cell vs. 548 +/- 105 / cell). In addition, we collected all of the data from cells with intensity levels <720, designed to mimic the distribution of the GS-FH1 cells (Fig. 6 supplement 3). We then compared the sarcomere metrics (sarcomere number, sarcomere length, sarcomere width) between the full data set and the two low intensity subsets:

• Sarcomere number is the only non-normal metric. We therefore used the Mann Whitney U test, which shows no difference between all 3 WT distributions.

• We compared Z-line lengths by one-way ANOVA and Tukey's post hoc tests, again finding no significant difference for all distributions.

• Sarcomere length shows a weakly significant difference (p=0.038) between the whole WT data set and bio rep 1, but no difference between the whole WT data set and the HA<720 group.

Thus, cells expressing wild type FHOD3L at levels comparable to levels detected in GS-FH1 mutant rescues, are fully rescued. Based on these findings we conclude that the expression levels in the GS-FH1 are high enough to rescue the FHOD3 knock down, supporting our conclusion that the defect is due to loss of elongation activity. We have added this analysis and discussion to the revised manuscript.

**Recommendations for the authors:**

**Reviewing Editor Comments:**
You will see that the 3 reviewers are very positive about your work and appreciate the elegant combination of biochemical assays and functional tests in cardiomyocytes. We've had a long discussion with them and we all agree that two experiments deserve further effort to make the conclusions of your paper more convincing.

Thank you.

The first experiment is the TIRF elongation assay, where the two biochemist Reviewers remain doubtful that these short events are really due to the presence of a formin at the end of the filament. One of them suggests that two-color imaging with a labeled formin should clearly prove this point.

We agree that the elongation assays can be improved. Given the similarity of processivity of Fhod3L, Fhod3S and *Drosophila* FhodA (measured by a distinct method), we are inclined to believe them. However, the reviewer raises an excellent point about the accuracy of the measurements given the resolution (and noise) of the data. We are interested in the two-color imaging assay but do not believe it will necessarily simplify the analysis. We suspect that Fhod spends more time at/near the barbed end than is apparent based on elongation rates. The fact that we see repeated events on individual filaments at such low concentrations of FHOD3L (0.1 nM) supports this idea. Otherwise, the likelihood of FHOD3L finding barbed ends so often is really quite low.

We will return to these experiments, using alternate methods, curious to see what else we learn. In the meantime, we conducted more thorough analysis, including controls, and improved visualization of example traces. Data for elongation analysis and kymographs were acquired with Jfilament. We stretched the x-axis (time) in kymographs for FHOD3L-CT (Fig. 2F), FHOD3S-CT (Fig. 2, supplement 2C), FHOD3L-CT K1193L (Fig. 3, supplement 1A), and actin alone (Fig 2G), and highlighted regions of analysis. The slopes for these regions, separated based on intensity, were fit to the data in KaleidaGraph. The fits are offset from the data such that they do not obscure the filaments and corresponding rates are given. The fact that we never see fast dim regions when FHOD3 is not present, as shown in Fig. 2H and that the frequency of dim events is markedly increased (Fig. 2-supplements 1G and 2E) give us confidence that the events are real. We acknowledge in the text that the precise values of the short events may be inaccurate due to the resolution of our experiments. We hope the reviewers are convinced by the improved analysis.

The second experiment is the sarcomere assembly defect phenotype in the GS-FH1 rescue condition. This requires further investigation, as the extremely low level of GS-FH1 signal in transfected cells in Figure 6A may reflect a failure of actin-binding/nucleation in vivo, rather than its inability to elongate F-actin. Although you show that GS-FH1 can bind to sarcomeres when they are present, this may be due to a lack of siRNA activity in these cells, such that endogenous FHOD3L is still present. In this possible scenario, GS-FH1 could dimerize with endogenous FHOD3L.

We agree that the sarcomeres we see are likely to be residual and could reflect some remaining endogenous FHOD3. The reviewers are concerned about the low protein levels in the GSFH1 rescues. First, we do not agree that the levels are “extremely” low. Through careful analysis, we established that 3xHA-FHOD3L intensities between 300 and 3000 a.u./um^2^ were sufficient for full rescue. The mean for the GSFH1 experiments is 533 +/- 93, which is well within this range. Furthermore, we did not observe correlation between sarcomere number, length, or width and HA intensity over the full range collected for wild type FHOD3L or within the GS-FH1 data. We previously showed pilot data but now show correlation analysis for every analyzed cell (Fig. 4 – figure supplement 1 D-F). We conducted this analysis on all of the mutant rescue experiments (Fig. 6-supplement 1). Finally, we identified two subpopulations of the wildtype rescue data. One is all of the cells with HA intensity < 720, which gives a distribution of mean 545 +/- 85. The second set is the first biological replicate of wild type rescue, which has a distribution of mean 560 +/- 160. Again correlation shows little relationship between HA levels and sarcomere metrics. Nevertheless, we show intensity level matched images in Fig 6, as opposed to images reflecting average intensities.

The critical question remains whether the phenotype is due to low protein levels or due to loss of elongation by FHOD3L. Notably, we now show a cell that is full of sarcomeres and has relatively high FHOD3L levels as well, consistent with available binding sites stabilizing mutant protein but not ruling out heterodimerization (Fig. 6 – figure supplement 2C). Others have expressed mutant FHOD3L in a wild type background in mice. They observed poisoning, consistent with heterodimerization. Thus, it is possible that, as suggested, the FHOD3L-GSFH1 detected in sarcomeres is in fact heterodimerized with residual endogenous FHOD3L. In this case, we would still conclude that the protein is not functional enough to rescue, supporting a role for the FH1 domain.

In the future, we plan to perform experiments with compromised, but not inactive, FH1 domains, as we discuss in the paper.

We hope that you will find these comments useful.

Yes, the comments were thoughtful and helped us write a better paper. Thank you.

**Reviewer #1 (Recommendations for the authors):**
Some experiments should be described and analyzed more carefully. This lack of clarity calls into question the interpretation of some experiments. Overall, this study is not yet as convincing as it should be.Main recommendations:(1) Formin elongation phases in the TIRF experiment are not convincing. They are rare and it is difficult to see any significant difference between the control movie without FHOD3L-CT and the movie with FHOD3L-CT. Filaments assembled in the absence of FHOD3L-CT also show some fluorescence inhomogeneity (which is normal), and measurements of formin elongation rates and capping times are not convincing (for example, the kymograph of the control profilin-actin situation in Figure 2F also shows a fast elongation phase on the right).

Please see response above. We conducted more thorough analysis and created improved visualizations. We hope the data are more convincing now.

It is also difficult to understand how an accurate measurement can be made from these noisy kymographs, and the method section should explain that precisely.

This is a valid point. We added details of analysis to the methods section and we discuss the fact that the measurements are at the limit of our resolution in the paper. We rely on the large (~3-fold) difference in elongation, more than specific elongation rates for our interpretation.

One of the problems is that these events are too transient to quantify well with noisy data. I noticed that the formin concentration used in these movies is quite low (0.1 nM FHOD3L-CT). Is there a reason for this? Is it possible to increase the formin concentration to increase the number of formin capping/elongation events and provide more convincing movies?

We acknowledge that the data are noisy. We felt that it was necessary to perform experiments with filaments only tethered at one end, leaving the growing end free. We did so, in part, because when we did experiments with biotinylated actin to anchor the filaments down, we observed pauses in the absence of formin. Ultimately, we compromised, using anchored seeds and a relatively low concentration of NEM-myosin to decrease motion of the actin filaments.

The experiments were performed with such low FHOD3L-CT because it was a potent nucleator in TIRF assays, making data analysis nearly impossible with more formin present. FHOD3S-CT and FHOD3L-CT K1193L behaved somewhat differently between these experiments and we were able to perform them with 1 nM formin.

Not seeing formin at the tip of the filaments is an additional difficulty because we do not know if these pauses occur because formin is stuck to the coverslips (which could very well happen with these sticky proteins) or freely bound at the end of a filament as the text suggests. Is there any argument in favor of one scenario over the other?

This will be an important experiment. As described above, we suspect that Fhod spends more time at/near the barbed end than is apparent based on elongation data. The fact that we see repeated events on individual filaments at such low concentrations of FHOD3L (0.1 nM) supports this idea. Otherwise, the likelihood of FHOD3L finding barbed ends so often is really quite low. In order to address the question about the cause of pauses, we reviewed our data, finding that 38 of 40 bursts were preceded by pauses. We do, however, discuss that we cannot rule out non-specific interactions with the surface.

(2) Pyrene elongation assays in the presence of profilin are actually more convincing to test the elongation ability of formins. However, such an assay is not presented for all mutants. It should be.

While we agree to some extent with this comment, we did not include the pyrene data for all of the mutants because the shapes of the curves were even more complicated than those seen with wild type FHOD3L-CT rendering them uninterpretable.

(3) Some experiments (e.g. in Figure 2E) are performed with yeast profilin, while others (e.g. in Figure 2F) are performed with human profilin. Obviously, both profilins could modulate formin activity differently and the side-by-side interpretation of both experiments is difficult. Could the authors stick to human profilin for all experiments?

We used to always perform pyrene assays with yeast profilin because it was known to be insensitive to pyrene. These data were collected before we realized that the affinity of human profilin for actin is so high that we could probably do everything with this profilin. We have compared the two profilins for other formins, e.g. Delphilin, Capu, and did not observe detectable differences.

Minor recommendations:(1) The pyrene assays with the light blue colored curve choice are not ideal. I have difficulties seeing some of the curves.

Thank you. We added symbols to a subset of the traces to make them more visible.

(2) In the same curves, I can't understand what the +3.75 and 0.078 numbers mean. Could these results be plotted in a clearer way?

These values are the lowest concentrations in the range tests. They were matching light blue with black outline for visibility. We added symbols and changed the color of the numbering for improved visibility/understanding.

(3) In Figure 2D, is the Kd of I1163A really determined only from 2 experimental data points?

Of course not. We now show the figure with extended axes in Fig. 2 - figure supplement 1C.

(4) In Figure 2C, the shape of the curves suggests that this is not a pure capping assay, but a mix of capping and nucleation. It's not dramatic but could lead to an under-estimation of the capping efficiency.

We agree with the reviewer that the complicated shapes confound interpretation. Our analysis is based on the earliest slopes, in part, for this reason. We added discussion of this complication to the text.

**Reviewer #3 (Recommendations for the authors):**
Suggestions for additional experiments:(1) To evaluate whether GS-FH1 alone can indeed interact with existing actin filaments in vivo, the authors may consider performing immunoprecipitation assays with GS-FH1 extracted from rescued NRVMs.

An IP of GS-FH1 from cells could show actin filament side binding but, unfortunately, will not provide any information about filament end binding, which is of much greater interest.

It will be helpful to show phalloidin staining in GS-FH1 rescues in a similar manner as in Figure 6-supplement 1, panel B, and compare that with mock rescue in Figure 4 panel D. It will be essential to prove this prior to concluding that actin elongation activity is essential for sarcomere assembly.

This is an excellent suggestion. We now include images of phalloidin stained cells from both K1193L and GS-FH1 rescues (Fig. 6A’ – supplement 2A,B). We were intrigued to see small actin punctae that were sometimes aligned. We speculate that these could be pre-premyofibrils and suggest that this is further evidence that the GS-FH1 protein is not completely unstable.

(2) Prior to sarcomere assembly, a-actinin is known to form short bundles with actin filaments (I-Z-I complex) without clearly defined periodicity. This semi-ordered state then transforms into the more ordered sarcomeres with periodic spacing. It will be valuable to show the phalloidin staining in addition to the a-actinin IF consistently across all conditions. This may lead to further insights into the defects of sarcomere assembly. Along the same vein, higher magnification images showcasing several sarcomeres will help the readers evaluate these defects.

We agree that there are additional valuable measurements to be made. In order to favor synchronized contraction, we plated the cells at too high a density to reliably identify IZI complexes. We have included some zoomed in images of the phalloidin staining.

Recommendations for improving the writing:The authors mentioned the interaction between cardiac MyBP-C and FHOD3L as essential for the localization of FHOD3L to the C-line of the sarcomere. Can they discuss whether this interaction is important for the role of FHOD3L in sarcomere assembly? If so, how?

This is a very interesting question that we cannot answer at this time.

Minor corrections to the text and figures:In the legend of Figure 2-Figure Supplement 1, the labels of (F) and (E) are swapped.

Thank you for catching this.